# Perceptron-as-Opinion-Dynamics (POD): A Unified and Interpretable Machine Learning Framework for Opinion Dynamics

## Abstract

We introduce **Perceptron-as-Opinion-Dynamics (POD)**, a dual-activation perceptron framework that learns opinion dynamics directly from data with full interpretability. Each parameter in POD corresponds to a concrete social mechanism, including agent-specific inertia, dynamic influence networks, persistent bias, and nonlinear modes of perception and expression. With appropriate parameter choices, POD exactly recovers canonical linear models such as DeGroot, Friedkin–Johnsen, and Altafini, closely approximates nonlinear models like Hegselmann–Krause, and naturally extends to kinetic and extremist cases. When trained end-to-end, POD generalizes beyond these settings, achieving up to 83% faster convergence and reducing prediction error by 40–82% compared to canonical baselines on real-world data. By unifying fragmented opinion dynamics models into a single trainable and interpretable neural framework, POD lays a robust foundation for modeling complex belief evolution in social systems, addressing challenges that have persisted for decades.

**Keywords:** Opinion dynamics, Social networks, Perceptron learning, Residual networks, Explainable AI

## 1 Introduction

Opinion dynamics models how beliefs and attitudes evolve within social networks, shaping collective outcomes such as elections, protests, and cultural shifts. Over decades, a wide variety of models have been proposed—each capturing a narrow mechanism such as consensus, stubbornness, bounded confidence, antagonism, or imitation. While influential, these models face long-standing limitations that constrain both their theoretical scope and practical utility.

First, the field remains fragmented and mutually irreproducible: each model isolates a single mechanism, and no unified update rule can replicate the dynamics of others. As a result, opinion dynamics has lacked a common foundation for decades. Second, most models rely on oversimplified linear averaging assumptions, while nonlinear behaviors (e.g., thresholds, saturation) are added in an ad hoc manner. Third, networks are typically assumed to be static or highly stylized, overlooking the fact that real-world influence structures evolve dynamically with events, social impressions, cultural shifts, or echo-chamber effects. Fourth, existing approaches face scalability trade-offs: kinetic PDE formulations lose agent-level interpretability, while agent-based simulations are difficult to scale. Fifth, many models fail to predict real-world opinion trajectories—largely because they cannot be trained directly on data. Finally, despite structural similarities to neural networks, opinion dynamics remains disconnected from machine learning: traditional models are not learnable, and ML approaches lack interpretability grounded in social theory.

We address these challenges by introducing **Perceptron-as-Opinion-Dynamics (POD)**—the first unified and trainable framework that bridges opinion dynamics and modern machine learning. POD generalizes canonical models—including consensus, stubbornness, bounded confidence, antagonism, and polarization—within a single parametric rule that features time-varying influence structures and heterogeneous agent inertia, while naturally extending to extremist and kinetic dynamics. Beyond unification, POD serves as a

novel interpretable ML building block: each parameter maps directly to a meaningful social mechanism and can be optimized end-to-end from data.

Our work includes the following key contributions:

(i) We present POD as the first framework in decades to unify and replicate all major classes of opinion dynamics, resolving long-standing theoretical fragmentation.

(ii) We introduce a dual-activation perceptron with interpretable, learnable parameters, including time-varying influence structures that require no prior knowledge of network topology.

(iii) We construct synthetic benchmarks and process real-world ICEWS geopolitical data by aggregating tone intensities into interpretable dyadic opinion scores.

(iv) Our empirical results show that POD exactly reproduces linear dynamics, approximates nonlinear dynamics with faster convergence, and substantially outperforms canonical baselines on real data.

These results establish POD as a unified, interpretable, and trainable framework for modeling opinion dynamics—and a solid foundation for future theoretical and empirical work.

## 2 Related Work

In opinion dynamics, linear consensus models established the foundations. The French–DeGroot process De-Groot (1974) models opinion exchange as repeated averaging with fixed influence matrices and guarantees convergence under network connectivity, but it cannot represent stubbornness or confidence thresholds. The Friedkin–Johnsen (FJ) model Friedkin & Johnsen (1990) introduced innate beliefs and persistent disagreement, extending consensus to stubborn agents but remaining linear with static network structures.

Nonlinear bounded confidence models advanced the field by introducing state-dependent influence. The Hegselmann–Krause (HK) model Hegselmann & Krause (2002) allows individuals to interact only with neighbors within a confidence interval, producing clusters and fragmentation through time-varying influence matrices that depend on current opinion states. The Deffuant–Weisbuch model Deffuant et al. (2000) implements pairwise compromise under thresholds, yielding gradual polarization. These formulations highlight clustering dynamics but cannot reproduce antagonism or persistent disagreement.

Antagonistic and signed network models extend consensus to capture polarization. Signed Degroot Altafini (2013) showed that signed influence matrices can drive groups toward bipartite consensus, explaining enemy–ally structures in social and geopolitical networks.

Discrete choice and statistical physics models provide complementary perspectives. The Voter model Clifford & Sudbury (1973) and Sznajd model Sznajd-Weron & Sznajd (2000) describe binary imitation through local copying or majority rules, closely related to Ising-like spin systems. Extremist variants Hegselmann & Krause (2015) anchor a subset of agents at extreme values, modeling radicalization. Kinetic and Boltzmann-type approaches Toscani (2006); Pareschi & Toscani (2013) treat interactions as collisions, producing PDE-level descriptions of opinion distributions. More recent nonlinear consensus models (e.g., Bizyaeva et al., 2020–22) introduced sigmoidal interaction functions, uncovering bifurcations, multistability, and abrupt transitions. These physics-inspired approaches capture rich behaviors but are computationally heavy and often lose interpretability at the agent level.

Attempts to connect opinion dynamics with machine learning have also emerged, but with limitations. Early work by Vicente, Martins, and Caticha Vicente et al. (2009) introduced a Boolean perceptron analogy to explain how consensus-seeking can paradoxically produce polarization. However, this was a theoretical construction rather than a trainable model, as it did not learn from data. While conceptually novel, this line is restricted to binary states and does not subsume averaging or bounded-confidence dynamics. More recently, sociologically-informed neural networks Okawa & Iwata (2022), probabilistic bounded confidence models Wang et al. (2024), stance-dynamics RNNs Zhu et al. (2020), and graph neural networks for influence prediction Fan et al. (2019); Qiu et al. (2018); Tran et al. (2023) have demonstrated empirical power on real

data. Yet these methods act as black boxes: they lack theoretical interpretability and cannot recover canonical opinion dynamics in closed form Wu et al. (2021); Zhang et al. (2022); Zhou et al. (2020).

ODNet Chaudhury et al. (2025) introduced a bounded-confidence mechanism into graph neural message passing, improving representation learning on graphs and hypergraphs. While ODNet demonstrates the promise of embedding opinion-dynamics principles in machine learning, it is limited to a single model family and serves primarily as a GNN architecture, whereas our proposed POD provides a unified, interpretable framework that subsumes multiple canonical models.

In contrast, the perceptron Rosenblatt (1958) and residual networks He et al. (2016)—two of the most fundamental neural architectures in machine learning—have not previously been applied to opinion dynamics. POD shows that both arise as strict subsets of its update rule, positioning it as the first framework to bridge decades of opinion dynamics research with modern machine learning while preserving interpretability.

## 3 Perceptron-as-Opinion Dynamics

### 3.1 Formulation

Let us define Perceptron-as-Opinion-Dynamics (POD) as the following parametric update rule for opinion evolution:

$$\boldsymbol{x}_{t+1} = \boldsymbol{\alpha} \odot \boldsymbol{x}_t + (1 - \boldsymbol{\alpha}) \odot \psi(\boldsymbol{W}_t \phi(\boldsymbol{x}_t) + \boldsymbol{b}) \tag{1}$$

or, equivalently at the agent level,

$$x_i^{t+1} = \alpha_i x_i^t + (1 - \alpha_i) \psi \left( \sum_{j=1}^{N} W_{ij}^t \phi(x_j^t) + b_i \right) \tag{2}$$

where $\boldsymbol{x}_t \in \mathbb{R}^N$ is the vector stacking the opinions of $N$ agents at time $t$ (with the $i$-th agent's opinion denoted by $x_i^t$). Each term in the update rule has a clear social interpretation. The inertia vector $\boldsymbol{\alpha} \in [0,1]^N$ encodes each agent's memory/stubbornness—each agent $i$ has its own inertia $\alpha_i$, allowing for heterogeneous stubbornness across the network. The time-varying influence matrix $\boldsymbol{W}_t$ encodes who listens to whom at each time $t$, allowing for dynamic network evolution: $W_{ij}^t$ represents how much agent $i$ listens to agent $j$ at time $t$. The influence matrix $\boldsymbol{W}_t$ may vary over time to reflect evolving social network structure, changing relationships, or state-dependent communication rules. The bias vector ($\boldsymbol{b}$) represents persistent exogenous forces such as media narratives or cultural baselines, which nudge opinions independently of peer influence. The inner activation function $\phi$ models how opinions are expressed or transmitted in communication—often subject to compression, simplification, or distortion. The outer activation function $\psi$ models how agents internalize social signals, capturing effects like saturation, thresholds, or conviction. This single update rule recovers consensus, stubbornness, bounded confidence, antagonism, and polarization as special cases, and extends naturally to extremist and kinetic formulations; with $\boldsymbol{\alpha} = \boldsymbol{0}$ and $\phi = \text{id}$, it reduces to a standard perceptron; with $\boldsymbol{\alpha} > \boldsymbol{0}$, it generalizes to a ResNet-like residual block with learnable intensity.[1]

This formulation captures three key aspects of real opinion dynamics: individuals retain partial memory of prior views ($\boldsymbol{\alpha} \odot \boldsymbol{x}_t$), social communication distorts or compresses information ($\phi$), and recipients interpret signals nonlinearly ($\psi$), all under persistent cultural or media bias ($\boldsymbol{b}$). These components yield a flexible yet interpretable description of opinion learning.

### 3.2 Opinion Dynamics Learning

Building on this parametric structure, we now describe how POD transitions from canonical models with fixed update rules (averaging, bounded confidence, antagonistic links) to a learnable, data-driven setting. Its parameters $\theta = \{\boldsymbol{W}_t, \boldsymbol{b}, \boldsymbol{\alpha}, \phi, \psi\}$ are optimized from observed trajectories rather than fixed by assumption.

---

[1]Notation: $\boldsymbol{\alpha}$ denotes the agent-wise inertia vector; $\boldsymbol{W}_t$ denotes the time-varying influence matrix at time $t$; $\boldsymbol{b}$ denotes the bias vector. $\phi$ and $\psi$ denote the inner and outer activation functions, respectively, which can be *linear* (identity) or *nonlinear* (e.g., sigmoid, tanh, ReLU, Leaky ReLU, SiLU, ELU).

In practice, POD is trained with a supervised k-step trajectory loss. Training data is chunked into $(\boldsymbol{x}_t, [\boldsymbol{x}_{t+1}, \ldots, \boldsymbol{x}_{t+k}])$ pairs, and the loss is:

$$\mathcal{L} = \frac{1}{B} \sum_{b=1}^{B} \frac{1}{k} \sum_{i=1}^{k} |\boldsymbol{x}_{t_b+i} - \hat{\boldsymbol{x}}_{t_b+i}|_2^2 \tag{3}$$

where $B$ is the batch size and $k$ is the prediction horizon. This encourages stable and accurate multi-step forecasts, improving performance over long time horizons. For $k = 1$, the loss reduces to a standard per-timestep prediction error.

Crucially, learnability does not sacrifice interpretability: each parameter has a direct social analogue. The inertia vector $\boldsymbol{\alpha}$ measures memory or resistance to change for each agent; because each agent can have its own inertia parameter $\alpha_i$, POD can flexibly model heterogeneity in stubbornness or resistance to change, matching well-established models like Friedkin–Johnsen. The time-varying matrix $\boldsymbol{W}_t$ captures the evolving network of influence, allowing the model to adapt to changing social structures and state-dependent communication patterns; $\boldsymbol{b}$ encodes persistent media or cultural biases; $\phi$ represents how opinions are expressed during communication; and $\psi$ determines how signals are internalized with thresholds or saturation. Thus POD combines the analytic clarity of opinion dynamics with the predictive power of trainable neural architectures.

Thus, having specified the trainable parameters, we now formalize the connection between POD and canonical opinion dynamics models, demonstrating its universality as detailed next.

### 3.3   POD Parametric Mappings

Since POD is fully interpretable, we can map canonical models into its parameter space. This establishes POD as a unifying framework that subsumes all major opinion dynamics families.

**Theorem 1 (Unification of Canonical Opinion Dynamics Families).**

Let $F_t$ denote any canonical discrete-time opinion dynamics operator representing consensus, stubbornness, bounded confidence, antagonism, polarization, or related families—potentially with time- or state-dependent update rules. Then, there exists a parameterization $(\boldsymbol{\alpha}, \phi, \psi, \boldsymbol{b}, \boldsymbol{W}_t)$ of POD such that, for any initial state, POD produces trajectories that exactly match those of $F_t$.

*Proof sketch.* For **linear models** (e.g., DeGroot, Friedkin–Johnsen, signed DeGroot/Altafini), set $\phi = \psi =$ id and tune $\boldsymbol{\alpha}$, $\boldsymbol{W}_t$, and $\boldsymbol{b}$ to match the linear dynamics of $F_t$. For **nonlinear models** (e.g., Hegselmann–Krause, Deffuant–Weisbuch), POD offers two approaches: (i) set $\phi = \psi =$ id and use time-varying $\boldsymbol{W}_t$ that updates based on opinion distances to enforce confidence thresholds, or (ii) select nonlinear $\phi$ or $\psi$ to enforce thresholds directly. For antagonistic or polarization dynamics, use signed or state-varying $\boldsymbol{W}_t$. Discrete-choice families (e.g., Voter, Sznajd, Ising) can be recovered by thresholded activations or stochastic sampling within POD. Thus, all major canonical families and their extensions admit a POD parameterization. $\square$

This unification result demonstrates that POD provides a single, trainable framework capable of reproducing decades of fragmented opinion dynamics research and flexibly extending to new model families. Table 1 illustrates mappings for the main models experimentally validated, with further examples given in Appendix A.

## 4   Experiments and Results

We evaluate POD in four stages: replication of other opinion models (Part I) using POD parametric mappings, learnability by training on synthetic data (Part II), activation function ablation study (Part III), and forecasting on real-world ICEWS dataset (Part IV), demonstrating POD's versatility across theoretical, synthetic, and real-world domains.

Table 1: Parameter mappings from canonical models to POD

| Model | Canonical Update | POD Mapping | Opinion Dynamics |
|---|---|---|---|
| DeGroot (1974) | $\boldsymbol{x}_{t+1} = \boldsymbol{W}\boldsymbol{x}_t$ | $\boldsymbol{\alpha} = \boldsymbol{0}$, $\phi = \psi = $ id, $\boldsymbol{b} = \boldsymbol{0}$, $\boldsymbol{W}_t = \boldsymbol{W}$ (static) | Linear consensus |
| FJ (1990) | $\boldsymbol{x}_{t+1} = \lambda \boldsymbol{W}\boldsymbol{x}_t + (I - \lambda)\boldsymbol{x}_0$ | $\boldsymbol{\alpha} = \boldsymbol{0}$, $\phi = \psi = $ id, $\boldsymbol{b} = (I-\lambda)\boldsymbol{x}_0$, $\boldsymbol{W}_t = \lambda\boldsymbol{W}$ (static) | Stubborn agents |
| HK (2002) | Bounded-confidence averaging | $\boldsymbol{\alpha} = \boldsymbol{0}$, $\phi = \psi = $ id, $\boldsymbol{b} = \boldsymbol{0}$, $\boldsymbol{W}_t$ time-varying, row-normalized within $\epsilon$ | Dynamic $\boldsymbol{W}_t$, async updates |
| Signed DeGroot (Altafini, 2013) | $\boldsymbol{x}_{t+1} = \boldsymbol{W}\boldsymbol{x}_t$, $\boldsymbol{W}$ signed | $\boldsymbol{\alpha} = \boldsymbol{0}$, $\phi = \psi = $ id, $\boldsymbol{b} = \boldsymbol{0}$, $\boldsymbol{W}_t = \boldsymbol{W}$ (static, signed) | Polarization |

## 4.1 Part I: Canonical Opinion Dynamics Replication

We evaluate POD across 18 scenarios combining 6 canonical models with 3 network sizes ($N \in \{50, 400, 1000\}$). The scenarios span four model families: (A3) DeGroot on Barabási-Albert scale-free networks with hub attenuation, (B3) Friedkin-Johnsen with heterogeneous stubbornness parameters, (C2/C3) Hegselmann-Krause with wide ($\varepsilon = 0.3$) and narrow ($\varepsilon = 0.12$) confidence bounds on complete graphs with 30% asynchronous updates, and (D2/D3) Signed DeGroot on 2- and 4-community stochastic block models. This design tests POD's ability to replicate linear consensus, bounded confidence clustering, and antagonistic dynamics across different network topologies and scales.

Table 2: Experimental configurations for canonical model replication

| Scenario | Model | Topology | Key Parameters | Initialization |
|---|---|---|---|---|
| A3 | DeGroot | Barabási-Albert | m=3, top 5% hubs ×0.5 | Uniform [-0.6,0.6] |
| B3 | Friedkin-Johnsen | Barabási-Albert | $\lambda \in [0.6, 0.9]$, m=3, top 5% hubs ×0.5 | Uniform [-0.8,0.8] |
| C2 | Hegselmann-Krause | Complete | $\varepsilon = 0.3$, async=0.3 | Uniform [-0.7,0.7] |
| C3 | Hegselmann-Krause | Complete | $\varepsilon = 0.12$, async=0.3 | Multimodal (5 peaks) |
| D2 | Signed DeGroot | 2-community SBM | 2 blocks, signed edges | Uniform [-0.6,0.6] |
| D3 | Signed DeGroot | 4-community SBM | 4 blocks, ally pairs | Multimodal (4 peaks) |

Simulations ran until convergence with tolerance $10^{-6}$ and patience 10. These trajectories are used as ground truths for POD to replicate. For canonical model replication, we set $\boldsymbol{\alpha} = \boldsymbol{0}$ (homogeneous zero inertia for all agents, i.e., $\alpha_i = 0$ for all $i$). For DeGroot, HK, and Signed DeGroot models, this directly maps to their standard formulations. For Friedkin-Johnsen, the stubbornness is captured through the bias term $\boldsymbol{b} = (I - \lambda)\boldsymbol{x}_0$ rather than through $\boldsymbol{\alpha}$, maintaining the standard FJ mapping.

We use static influence matrices ($\boldsymbol{W}_t = \boldsymbol{W}$) for DeGroot, FJ, and Signed DeGroot models, while HK employs time-varying $\boldsymbol{W}_t$ to model evolving social ties. For HK, $\boldsymbol{W}_t$ is updated dynamically: neighbors $j$ are retained only if $|x_i^t - x_j^t| \leq \epsilon$, with rows renormalized. Asynchronous updates are modeled by

$$\boldsymbol{W}_t^{(\text{async})} = M_t \odot \boldsymbol{W}_t, \quad \mathbb{E}[M_t] = 0.3 \tag{4}$$

where $M_t \in \{0, 1\}^{N \times N}$ is a random binary mask ensuring that, on average, only 30% of agents update their opinions at each step, matching the async fraction parameter in the C2/C3 scenarios.

**Results.** POD replicates DeGroot, FJ, and Signed DeGroot at machine precision (RMSE $< 10^{-8}$). For HK, POD approximates bounded-confidence clustering with RMSE 0.04–0.15 while preserving structural statistics (clusters, polarization variance, signed assortativity). Convergence matches linear models and is

**52–83% faster** for HK. These results demonstrate that POD exactly recovers linear and signed models and closely approximates nonlinear HK while maintaining structural properties.

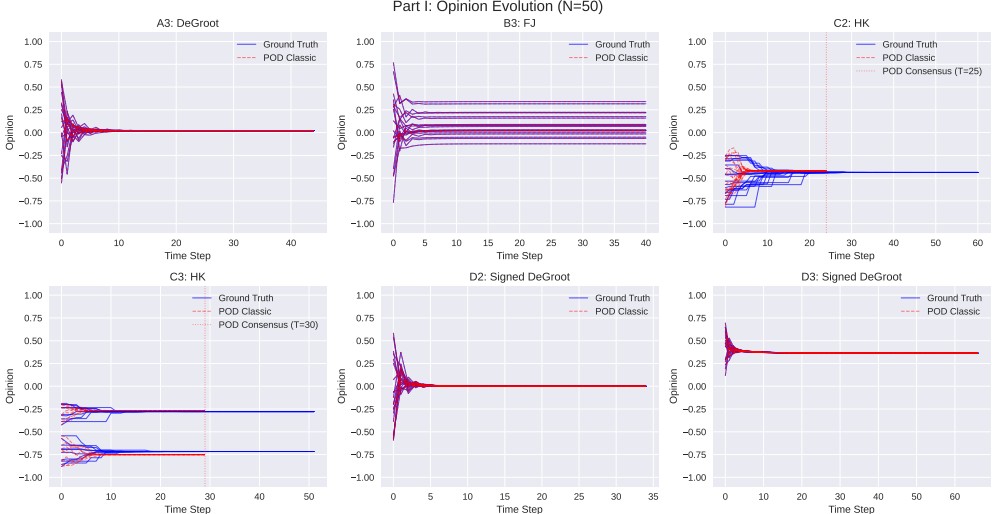

Figure 1: Replication of canonical dynamics by POD at $N = 50$ (DeGroot, FJ, Signed DeGroot, HK)

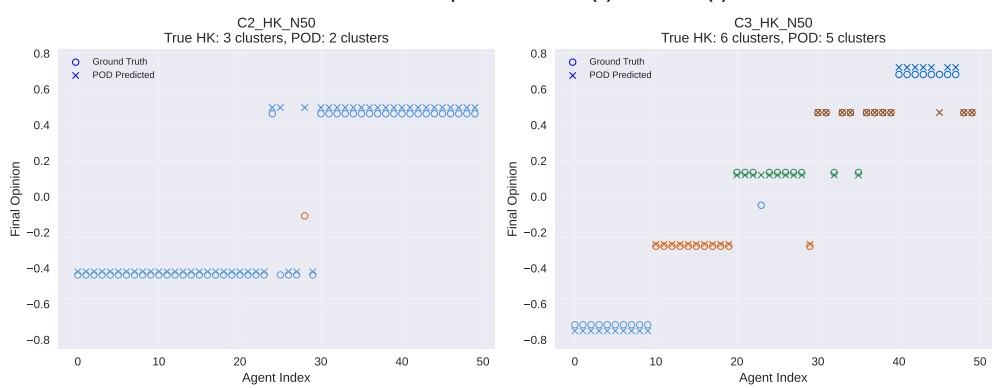

Figure 2: Clustering dynamics under Hegselmann–Krause model at $N = 50$

Detailed opinion trajectory visualizations for larger network sizes ($N \in \{400, 1000\}$) are provided in Appendix B.

## 4.2 Part II: Learnable POD

We focus on the most challenging scenario: Hegselmann–Krause (C3, $N = 1000$), while Appendix C contains results for all 18 scenarios. Each trajectory is split into early (5%), training (80%), validation (10%), and late (5%) segments, with shuffling for autoregressive training. This split tests POD both during early, dynamic opinion change and late-stage convergence and clustering.

In this setup, $\boldsymbol{W}_t$ is randomly initialized and updated with 70% dropout per step, mimicking HK's 30% asynchronous updates and requiring POD to infer both network topology and nonlinear dynamics from data. We use homogeneous (scalar) inertia, as in canonical models.

---

[2]Replication Quality: Perfect (RMSE $< 10^{-6}$), Very Good (RMSE $< 0.1$), Good (RMSE $< 0.2$). Performance: $+\%$ means POD faster convergence, $-\%$ means slower convergence compared to canonical model. $T$ denotes the time step to converge.

Table 3: Replication accuracy of POD across canonical models[2]

| Scenario | Model | POD Configuration ($\boldsymbol{W}_t = \boldsymbol{W}_{\mathrm{GT}}$, $\phi = \mathrm{id}, \psi = \mathrm{id}$, $\boldsymbol{\alpha} = \boldsymbol{0}$, $\boldsymbol{b} = 0$, if not specified) | $N$ | RMSE | Replication Quality | $T_{\mathrm{Ref}}$ | $T_{\mathrm{POD}}$ | Perf. |
|---|---|---|---|---|---|---|---|---|
| A3 | DG | — | 50 | 2.94e-09 | Perfect | 35 | 35 | 0.0% |
| A3 | DG | — | 400 | 3.90e-09 | Perfect | 41 | 41 | 0.0% |
| A3 | DG | — | 1000 | 2.13e-09 | Perfect | 40 | 40 | 0.0% |
| B3 | FJ | $\boldsymbol{b} = (I - \lambda)\boldsymbol{x}_0^{\mathrm{GT}}$ | 50 | 1.36e-08 | Perfect | 31 | 31 | 0.0% |
| B3 | FJ | $\boldsymbol{b} = (I - \lambda)\boldsymbol{x}_0^{\mathrm{GT}}$ | 400 | 1.16e-08 | Perfect | 32 | 32 | 0.0% |
| B3 | FJ | $\boldsymbol{b} = (I - \lambda)\boldsymbol{x}_0^{\mathrm{GT}}$ | 1000 | 1.10e-08 | Perfect | 28 | 28 | 0.0% |
| C2 | HK | Dynamic $\boldsymbol{W}_t$, $\varepsilon = 0.3$, async=0.3 | 50 | 0.1502 | Good | 49 | 15 | +69.4% |
| C2 | HK | Dynamic $\boldsymbol{W}_t$, $\varepsilon = 0.3$, async=0.3 | 400 | 0.1464 | Good | 58 | 14 | +75.9% |
| C2 | HK | Dynamic $\boldsymbol{W}_t$, $\varepsilon = 0.3$, async=0.3 | 1000 | 0.1346 | Good | 63 | 12 | +81.0% |
| C3 | HK | Dynamic $\boldsymbol{W}_t$, $\varepsilon = 0.12$, async=0.3 | 50 | 0.0443 | Very Good | 42 | 20 | +52.4% |
| C3 | HK | Dynamic $\boldsymbol{W}_t$, $\varepsilon = 0.12$, async=0.3 | 400 | 0.0672 | Very Good | 101 | 17 | +83.2% |
| C3 | HK | Dynamic $\boldsymbol{W}_t$, $\varepsilon = 0.12$, async=0.3 | 1000 | 0.0521 | Very Good | 60 | 28 | +53.3% |
| D2 | Signed DG | — | 50 | 1.94e-09 | Perfect | 25 | 25 | 0.0% |
| D2 | Signed DG | — | 400 | 1.49e-09 | Perfect | 27 | 27 | 0.0% |
| D2 | Signed DG | — | 1000 | 1.75e-09 | Perfect | 28 | 28 | 0.0% |
| D3 | Signed DG | — | 50 | 2.66e-08 | Perfect | 57 | 57 | 0.0% |
| D3 | Signed DG | — | 400 | 2.00e-08 | Perfect | 35 | 35 | 0.0% |
| D3 | Signed DG | — | 1000 | 1.77e-08 | Perfect | 28 | 28 | 0.0% |

POD is trained with one-step MSE loss ($k = 1$), batch size 1, $\phi = \mathrm{SiLU}$ for opinion evolution, and $\psi = \tanh$ for agent perception:

$$\mathcal{L} = \frac{1}{T} \sum_{t=1}^{T} \|\boldsymbol{x}_{t+1} - \hat{\boldsymbol{x}}_{t+1}\|_2^2 \tag{5}$$

Optimization uses Adam (rate $5 \times 10^{-3}$, decayed to $10^{-6}$), early stopping (patience 50), and dropout.

**Results.** For C3 ($N = 1000$), POD reaches training loss 0.0181, validation loss 0.0052, RMSE 0.045 (early), 0.006 (late), and rollout RMSE 0.023, recovering all six HK consensus clusters in the final stage (see Fig. 3). Since POD forms clusters more rapidly than HK, higher early-stage RMSE marks faster convergence, not weak prediction.

We see that learned parameters align with HK theory: sparse $\boldsymbol{W}_t$ (mean $0.001 \pm 0.003$) signals within-cluster influence only; $\alpha = 0.288$ balances memory and social influence; and nonzero $\boldsymbol{b}$ reflects persistent group effects that agents within clusters maintain similar opinions. POD thus learns bounded-confidence dynamics directly from limited, noisy data and does not require network knowledge.

Although we use homogeneous $\alpha$ here for comparability, POD supports heterogeneous $\boldsymbol{\alpha}$ for agent-specific inertia. The framework also supports any (linear or nonlinear) activation functions for real-world applications.

### 4.3 Part III: Activation Function Ablation Study

To systematically evaluate the impact of different activation function combinations on POD's performance, we conducted an ablation study on the real-world ICEWS dataset, which captures geopolitical interactions over time. After preprocessing, the dataset contains 251 active countries exchanging dyadic event-based interactions on a monthly basis across 337 time steps ( 28 years). Each dyadic interaction was aggregated into a tone score and rescaled to $[-1, 1]$, yielding a dynamic opinion network suitable for multi-agent forecasting.

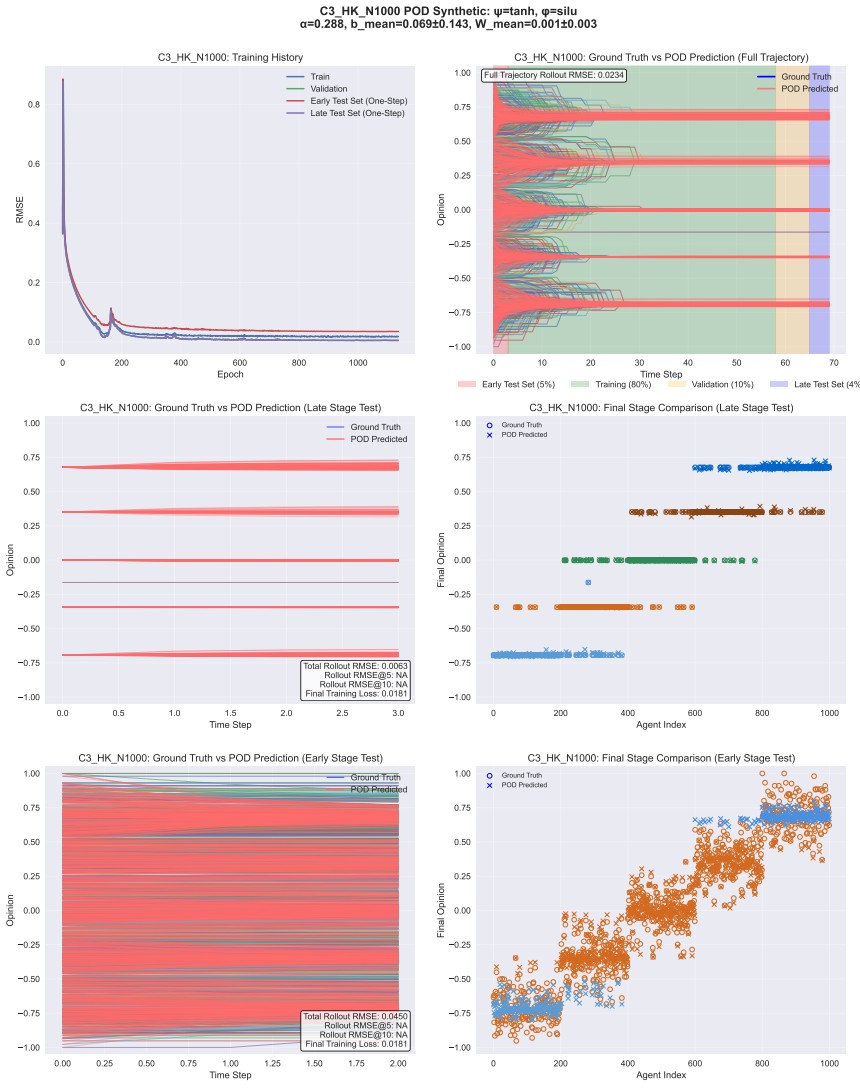

Figure 3: POD learning of Hegselmann–Krause dynamics ($N = 1000$)

We tested six key configurations: (1) $\psi = \tanh, \phi = \text{SiLU}$ with learnable vector $\boldsymbol{\alpha}$, (2) $\psi = \tanh, \phi = \tanh$ with learnable vector $\boldsymbol{\alpha}$, (3) $\psi = \tanh, \phi = \text{Leaky ReLU}$ with learnable vector $\boldsymbol{\alpha}$, (4) $\psi = \tanh, \phi = \text{ELU}$ with learnable vector $\boldsymbol{\alpha}$, (5) $\psi = \text{id}, \phi = \text{id}$ with learnable vector $\boldsymbol{\alpha}$, and (6) $\psi = \tanh, \phi = \text{id}$ with fixed $\alpha = 0$ (vanilla perceptron baseline). This design allows us to isolate the contributions of different activation functions and the learnable inertia mechanism.

Table 4 presents the results across all configurations. The modern activation functions—SiLU (RMSE 0.148), Leaky ReLU (RMSE 0.149), and ELU (RMSE 0.150)—achieve nearly identical performance, with differences of only 0.001-0.002 RMSE. This indicates that POD is robust to the exact choice among these smooth, saturating activation functions. In contrast, canonical approaches (tanh-tanh, identity-identity, and vanilla perceptron) show noticeably worse performance with RMSE values of 0.178-0.179, representing approximately 20% relative increase in error compared to modern activations. This suggests that the key distinction is between modern and canonical activation functions rather than among the modern choices themselves. The vanilla perceptron baseline ($\alpha = 0$) performs worst, confirming the importance of learnable inertia for opinion dynamics modeling.

Table 4: Effect of activation functions on POD prediction performance.

| Configuration | $\psi$ | $\phi$ | Full Trajectory RMSE | Avg. 1-step RMSE | Avg. 5-step RMSE |
|---|---|---|---|---|---|
| SiLU | tanh | SiLU | **0.148** | **0.136** | **0.142** |
| Leaky ReLU | tanh | Leaky ReLU | 0.149 | 0.146 | 0.170 |
| ELU | tanh | ELU | 0.150 | 0.142 | 0.167 |
| Tanh-Tanh | tanh | tanh | 0.179 | 0.142 | 0.166 |
| Identity-Identity | id | id | 0.178 | 0.143 | 0.164 |
| Vanilla Perceptron | tanh | id | 0.179 | 0.143 | 0.164 |

Based on these results, we select the SiLU configuration ($\psi = \tanh, \phi = \text{SiLU}$) as our representative candidate for detailed comparison with canonical opinion dynamics models. While SiLU achieves the best performance, the negligible differences among modern activation functions suggest that interpretability, stability, and practical considerations can guide activation choice within this set, since their performance is nearly identical in practice.

### 4.4 Part IV: Real-World ICEWS Forecasting

To evaluate POD's real-world utility, we tested our best-performing SiLU configuration on the ICEWS dataset for comprehensive forecasting evaluation. For this experiment, we initialized $\boldsymbol{W}_t$ randomly (row-normalized) and trained POD end-to-end from data using Adam (learning rate $5 \times 10^{-4} \to 10^{-6}$), early stopping (patience = 200), and dropout = 0.1. We used our representative configuration: $\psi = \tanh$ and $\phi = \text{SiLU}$ with learnable vector $\boldsymbol{\alpha}$. The data is split as 80% training, 10% validation, and 10% late testing. For this real-world experiment, we used vector $\boldsymbol{\alpha}$ to capture heterogeneous agent-specific inertia, allowing the influence matrix $\boldsymbol{W}_t$ to evolve dynamically to capture changing geopolitical relationships.

The training objective used a k-step trajectory loss to promote multi-step forecasting stability:

$$\mathcal{L}_{\text{train}} = \frac{1}{B} \sum_{b=1}^{B} \frac{1}{k} \sum_{i=1}^{k} \|\boldsymbol{x}_{t_b+i} - \hat{\boldsymbol{x}}_{t_b+i}\|_2^2$$

Where $B = 4$ is the batch size and $k = 5$ is the rollout horizon. This choice of k balances one-step accuracy and long-horizon generalization, aligning with the slow-changing nature of geopolitical opinion dynamics.

**Results.** POD achieved strong performance on the ICEWS dataset with a full trajectory RMSE of 0.148, significantly outperforming all canonical baselines. The learned parameters show meaningful patterns: $\boldsymbol{\alpha}_{\text{mean}} = 0.018 \pm 0.032$ indicates agents learned to be highly responsive to social influence, while the sparse influence matrix $\boldsymbol{W}_t$ (mean $0.004 \pm 0.014$) reflects the selective nature of geopolitical interactions.

Table 5 presents a comprehensive comparison between POD and canonical opinion dynamics models on the ICEWS dataset. POD demonstrates substantial improvements over all baselines: 82.4% improvement over DeGroot, 82.8% over Signed DeGroot, 82.3% over FJ, and 40.4% over HK. These results validate POD's capacity to learn complex, time-varying social dynamics directly from real-world data without requiring prior knowledge of network topology.

However, analysis of the forecasting results reveals important limitations. While POD excels at short-term prediction accuracy, it shows reduced generalization capability over longer horizons, particularly for outlier countries with unusual geopolitical behavior patterns. This limitation can be attributed to the inherent complexity of real-world international relations data, which contains significantly more noise, nonlinear interactions, and temporal dependencies than synthetic datasets. The limited training data (337 time steps for 251 countries) constrains POD's ability to learn robust long-term patterns, especially for countries with infrequent or atypical diplomatic interactions.

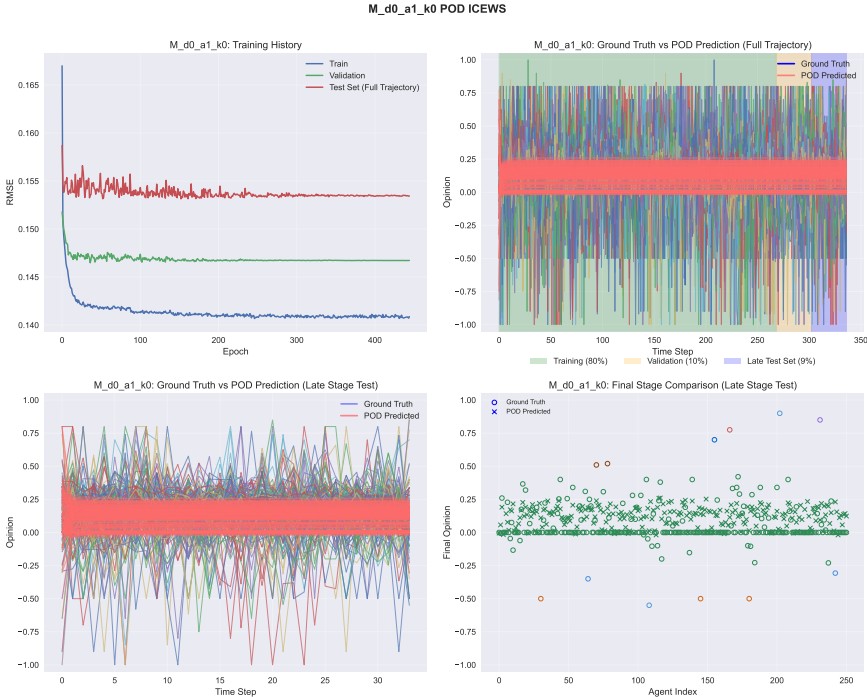

Figure 4: Forecasting belief evolution on ICEWS: POD vs baselines

Table 5: Performance comparison on ICEWS real-world dataset

| Model | Full Traj. RMSE | Avg. 1-step RMSE | Avg. 5-step RMSE | Avg. 10-step RMSE | Improvement |
|---|---|---|---|---|---|
| **POD (SiLU)** | **0.148** | **0.136** | **0.142** | **0.145** | - |
| HK | 0.249 | 0.189 | 0.190 | 0.189 | 40.4% |
| FJ | 0.835 | 0.786 | 0.824 | 0.831 | 82.3% |
| DeGroot | 0.842 | 0.794 | 0.832 | 0.839 | 82.4% |
| Signed DeGroot | 0.863 | 0.825 | 0.855 | 0.817 | 82.8% |

These results demonstrate POD's superior ability to capture the complex, nonlinear dynamics present in real-world opinion evolution over short to medium horizons, particularly the time-varying influence structures and heterogeneous agent behaviors that characterize geopolitical interactions. The framework's interpretable parameters provide valuable insights into learned social mechanisms, while its limitations highlight the challenges of modeling complex real-world systems with limited data.

## 5    Conclusion

We presented **Perceptron-as-Opinion-Dynamics (POD)**, a unified and interpretable framework that bridges canonical opinion dynamics with modern machine learning. By combining heterogeneous inertia, time-varying influence, social bias, and nonlinear activations within a dual-activation perceptron, POD recovers and generalizes a wide range of canonical models—including consensus, stubbornness, bounded confidence, and polarization—while remaining fully trainable from data.

Across extensive experiments, POD consistently demonstrated both theoretical fidelity and empirical strength. It exactly reproduces linear models, approximates nonlinear dynamics with faster convergence, and outperforms canonical baselines by up to 82.8% on real-world forecasting tasks. POD also offers inter-

pretability by design with each parameter mapped directly to a social mechanism, making it a transparent primitive for learning opinion dynamics from data.

While these results are promising, challenges remain. POD's long-horizon generalization is limited in data-scarce regimes, and modeling outlier agents with complex behavior remains difficult. These limitations reflect the inherent complexity of real-world social systems and point toward the need for richer data and adaptive learning strategies.

In conclusion, POD lays a foundation for future work at the intersection of social modeling and machine learning. We envision extensions to multimodal data, large-scale dynamic networks, and applications in policy, education, and collective behavior modeling—guided by interpretability and grounded in theory.

**Broader Impact Statement**

This work introduces POD—the first unified and trainable framework for opinion dynamics. By recovering decades of canonical models while extending them through data-driven learning, POD has the potential to reshape how researchers and practitioners study social influence. Its interpretability ensures that each learned parameter—stubbornness, bias, or network ties—can be linked back to clear social mechanisms. This allows POD to serve as both a forecasting tool and a theoretical lens for understanding consensus formation, polarization, and clustering in real-world systems.

However, the same properties that make POD powerful also raise risks. A model capable of learning evolving influence networks could be used for political manipulation, micro-targeted persuasion, or reinforcing societal biases embedded in data. While transparency helps to expose these mechanisms, it does not guarantee ethical use.

To mitigate these risks, we recommend three safeguards: (i) careful auditing of datasets and network structures for representativity and fairness, (ii) adoption of privacy-preserving practices when applying POD to human-derived data, and (iii) transparent reporting of model limitations and uncertainties. Responsible deployment should involve collaboration with social scientists and domain experts, especially in adversarial or high-stakes settings such as geopolitics.

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

Table 6: Extended parameter mappings for major opinion dynamics models

| Model | Canonical Update | POD Mapping | Opinion Dynamics |
|---|---|---|---|
| DeGroot (1974) | $\boldsymbol{x}_{t+1} = \boldsymbol{W}\boldsymbol{x}_t$ | $\alpha = 0$, $\phi = \psi = $ id, $b = 0$, $W_t = W$ (static) | Linear consensus |
| Friedkin–Johnsen (1990) | $\boldsymbol{x}_{t+1} = \lambda W \boldsymbol{x}_t + (I - \lambda) x_0$ | $\alpha = 0$, $\phi = \psi = $ id, $b = (I - \lambda) x_0$, $W_t = \lambda W$ (static) | Stubborn agents, innate beliefs |
| Hegselmann–Krause (2002) | Bounded-confidence averaging | $\alpha = 0$, $\phi = \psi = $ id, $b = 0$, $W_t$ time-varying, row-normalized within $\epsilon$ | Bounded confidence, clustering |
| Signed DeGroot (Altafini, 2013) | $\boldsymbol{x}_{t+1} = W \boldsymbol{x}_t$, $W$ signed | $\alpha = 0$, $\phi = \psi = $ id, $b = 0$, $W_t = W$ (static, signed) | Polarization, antagonism |
| Voter Model (Clifford & Sudbury, 1973) | $x_i^{t+1} = x_j^t$, $j$ random neighbor of $i$ | $\alpha = 0$, $\phi = $ id, $\psi = $ sign, $b = 0$, $W_t$ binary, static or dynamic | Binary imitation, majority rule |
| Sznajd Model (2000) | Majority update in pairs/groups | $\alpha = 0$, $\phi = \psi = $ sign, $b = 0$, $W_t$ local, dynamic | Group influence, logic rules |
| Lenz–Ising Model (1925)Ising (1925) | $x_i^{t+1} = \text{sign}(\sum_j J_{ij} x_j^t + b_i)$ | $\alpha = 0$, $\phi = $ id, $\psi = $ sign, $b = b$, $W_t = J$ (static) | Discrete states, field/bias |
| Kinetic/Boltzmann (Toscani, 2006) | Population-level, PDE: $f(x,t)$, via collisional kernel | $\alpha = 0$, nonlinear $\phi$ or $\psi$, $b = 0$, $W_t$ kernel/mean-field | Collisions, density-based |
| Extremist/Anchored Agent (2000s) | Some $x_i$ fixed at extremes | $\alpha_i = 1$ for extremists; others $0 \le \alpha_i < 1$; $b$ anchors as needed | Partial inertia, stubborn nodes |
| Nonlinear Consensus (2000s+) | $\boldsymbol{x}_{t+1} = W_t \psi(\boldsymbol{x}_t)$ or $\psi(W_t \boldsymbol{x}_t)$ | $\alpha = 0$, $\phi = $ id, $\psi$ nonlinear (e.g. tanh, threshold), $b = 0$, $W_t$ dynamic/static | Sigmoidal or threshold response |

## A  Extended Mappings of Canonical Models into POD

Table 6 provides comprehensive POD parametric mappings for major opinion dynamics models beyond those tested in our experiments, underscoring the framework's broad applicability across diverse social influence mechanisms.

## B  Additional Trajectory Visualizations across Scales

This appendix supplements Section 4.1 with trajectory visualizations for larger networks ($N = 400$ and $N = 1000$). The results demonstrate that POD maintains high replication fidelity across a range of dynamics—including consensus, stubbornness, bounded confidence, and polarization—even as network size increases. Combined with the $N = 50$ results in the main text, these figures provide a comprehensive view of POD's scalability and robustness.

Figures 5 and 6 illustrate POD's ability to exactly replicate canonical models across medium and large-scale networks. The trajectory comparisons show perfect alignment for linear models (DeGroot, FJ, Signed DeG-

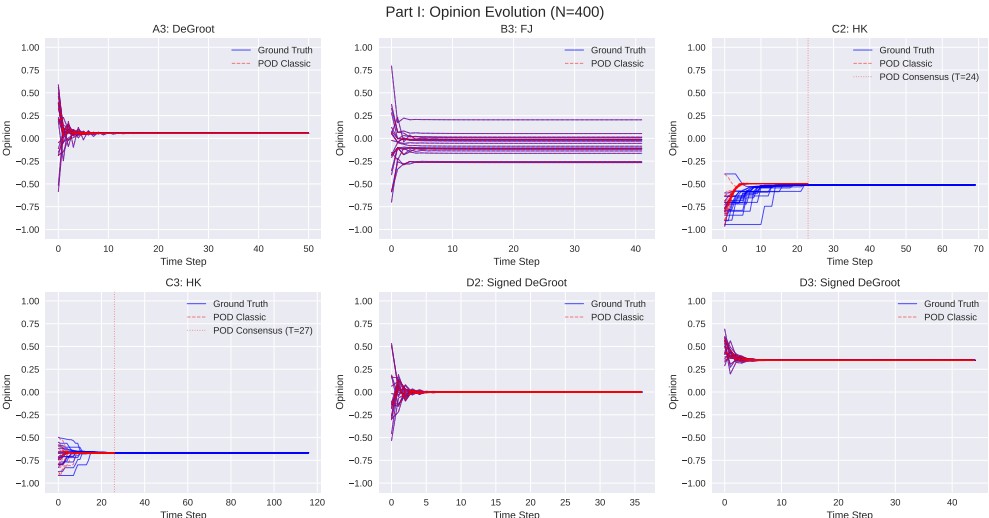

Figure 5: Replication of canonical dynamics by POD at $N = 400$

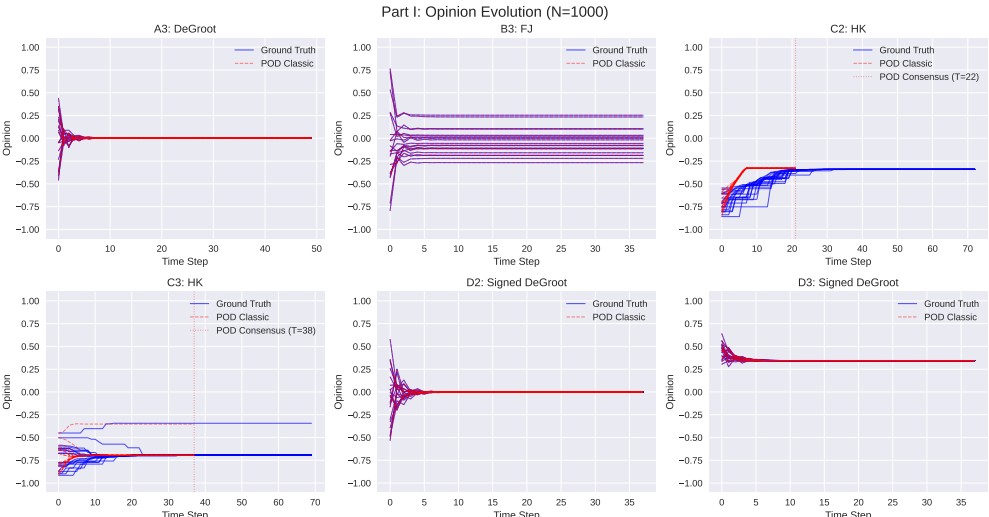

Figure 6: Replication of canonical dynamics by POD at $N = 1000$

root) and close approximation for nonlinear HK dynamics, demonstrating POD's scalability from hundreds to thousands of agents.

Figures 7–10 provide detailed analysis of convergence patterns and clustering behavior. Mean opinion evolution shows consistent convergence across network sizes, while HK clustering visualizations reveal POD's ability to capture complex bounded confidence dynamics, including multi-cluster formation and opinion fragmentation patterns that scale effectively with network size.

## C    Full Results for Learnable POD Experiments

This appendix presents the complete results from the learnable POD experiments across all 18 benchmark scenarios. Table 7 and the accompanying Figures 11–16 illustrate POD's ability to learn a broad spectrum of opinion dynamics directly from data—including linear consensus (DeGroot), stubborn agent dynam-

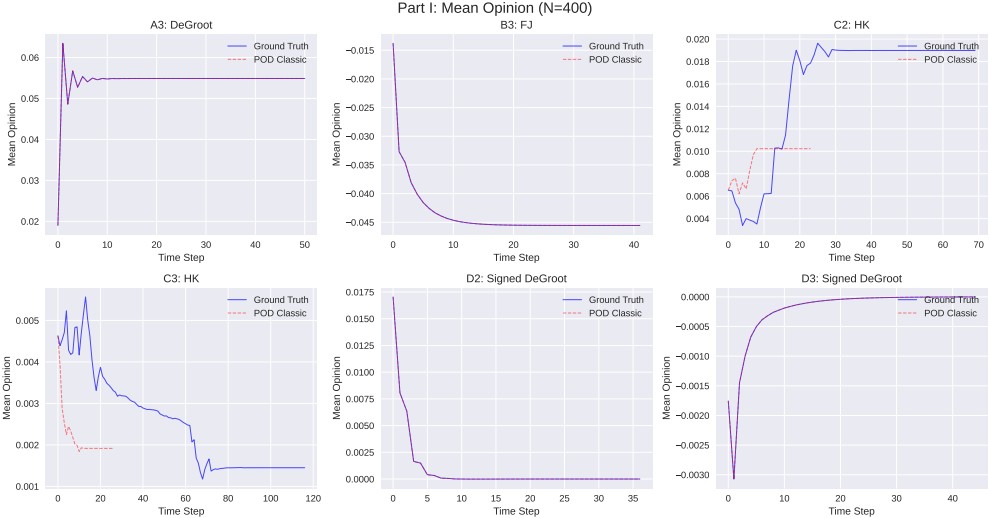

Figure 7: Scalability of POD replication across network sizes ($N = 400$)

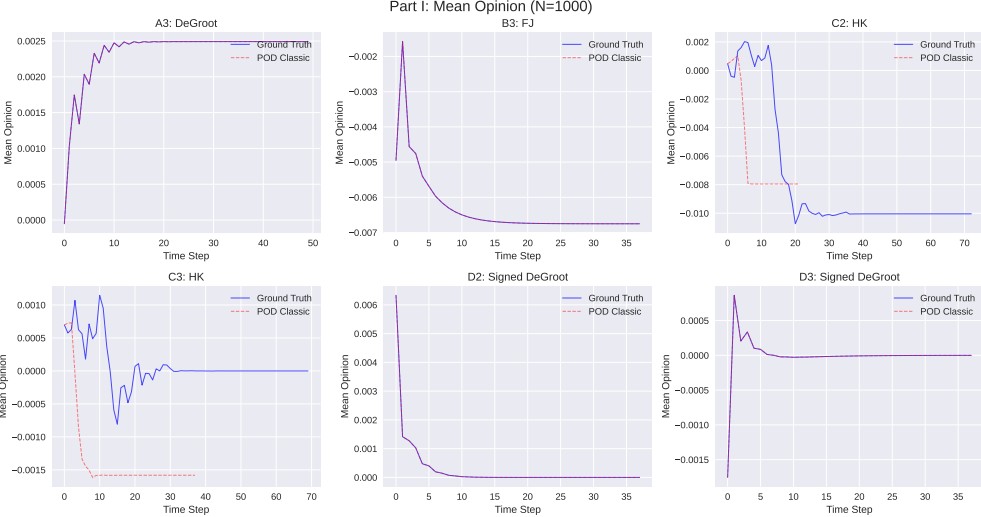

Figure 8: Scalability of POD replication across network sizes ($N = 1000$)

ics (Friedkin–Johnsen), bounded confidence clustering (Hegselmann–Krause), and antagonistic polarization (Signed DeGroot)—without prior knowledge of the underlying network.

Except for scenario C3 (HK, $N = 1000$), which is shown in the main text, this appendix includes detailed visualizations for all remaining cases. The results demonstrate that POD consistently captures the dynamics across diverse model families, parameter regimes, and network sizes.

Linear models (DeGroot, FJ) are exactly recovered with stable convergence across scales, while nonlinear models (HK) show successful approximation of complex clustering and fragmentation behaviors. In signed networks, POD learns antagonistic interactions and multi-polarization effects, scaling robustly from small ($N = 50$) to large networks ($N = 1000$), thus reinforcing its effectiveness as a general-purpose learning framework for opinion dynamics.

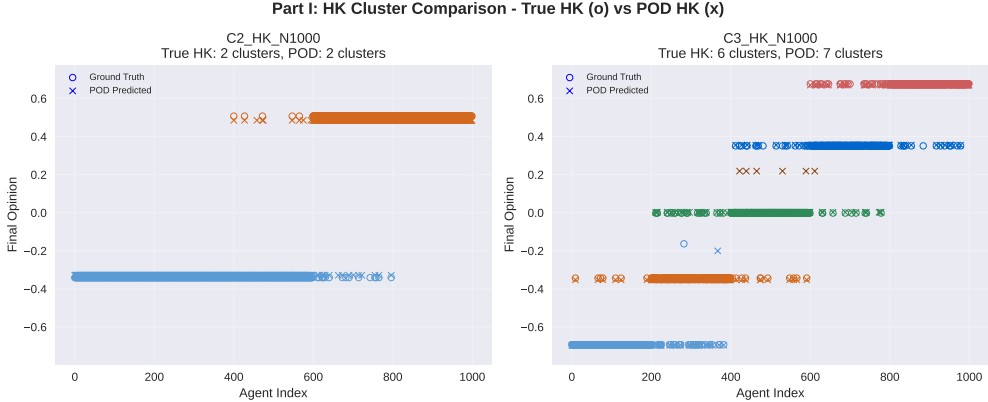

Figure 9: Clustering dynamics under Hegselmann–Krause model at $N = 400$

Figure 10: Clustering dynamics under Hegselmann–Krause model at $N = 1000$

Table 7: Comprehensive learning results across all experimental scenarios

| Scenario | Train RMSE | Val RMSE | Full RMSE | $\alpha$ | $b$ mean | Clusters | Epochs |
|---|---|---|---|---|---|---|---|
| A3 DeGroot ($N = 1000$) | 0.0053 | 0.0013 | 0.0181 | 0.022 | 0.001 | 1/1 | 2000 |
| A3 DeGroot ($N = 400$) | 0.0081 | 0.0014 | 0.0273 | 0.122 | 0.027 | 1/1 | 618 |
| A3 DeGroot ($N = 50$) | 0.0060 | 0.0008 | 0.0194 | 0.060 | 0.011 | 1/1 | 753 |
| B3 FJ ($N = 1000$) | 0.0079 | 0.0028 | 0.0382 | 0.212 | -0.003 | 1/1 | 1137 |
| B3 FJ ($N = 400$) | 0.0066 | 0.0023 | 0.0256 | 0.184 | -0.017 | 1/1 | 1033 |
| B3 FJ ($N = 50$) | 0.0074 | 0.0012 | 0.0307 | 0.314 | 0.068 | 1/1 | 937 |
| C2 HK ($N = 1000$) | 0.0258 | 0.0041 | 0.0460 | 0.281 | 0.046 | 2/2 | 1124 |
| C2 HK ($N = 400$) | 0.0329 | 0.0041 | 0.0458 | 0.273 | 0.005 | 2/2 | 995 |
| C2 HK ($N = 50$) | 0.0175 | 0.0012 | 0.0356 | 0.861 | 0.005 | 3/3 | 1482 |
| C3 HK ($N = 1000$) | 0.0181 | 0.0052 | 0.0234 | 0.288 | 0.069 | 6/6 | 1137 |
| C3 HK ($N = 400$) | 0.0193 | 0.0098 | 0.0269 | 0.350 | 0.066 | 6/6 | 927 |
| C3 HK ($N = 50$) | 0.0474 | 0.0091 | 0.0211 | 0.204 | 0.011 | 6/6 | 668 |
| D2 Signed DeGroot ($N = 1000$) | 0.0047 | 0.0010 | 0.0144 | 0.021 | 0.000 | 1/1 | 2000 |
| D2 Signed DeGroot ($N = 400$) | 0.0044 | 0.0009 | 0.0143 | 0.038 | 0.000 | 1/1 | 1317 |
| D2 Signed DeGroot ($N = 50$) | 0.0037 | 0.0006 | 0.0109 | 0.040 | 0.000 | 1/1 | 915 |
| D3 Signed DeGroot ($N = 1000$) | 0.0112 | 0.0011 | 0.0125 | 0.254 | -0.007 | 2/2 | 892 |
| D3 Signed DeGroot ($N = 400$) | 0.0175 | 0.0009 | 0.0118 | 0.218 | -0.015 | 2/2 | 860 |
| D3 Signed DeGroot ($N = 50$) | 0.0398 | 0.0034 | 0.0113 | 0.221 | 0.003 | 3/3 | 699 |

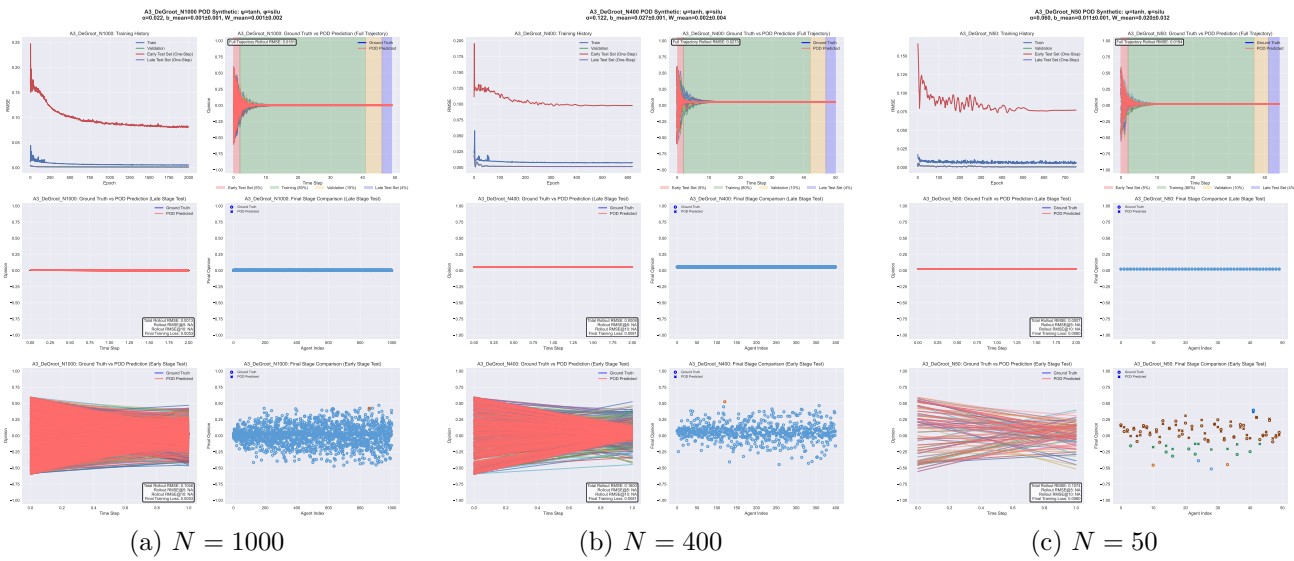

Figure 11: POD learning of DeGroot dynamics

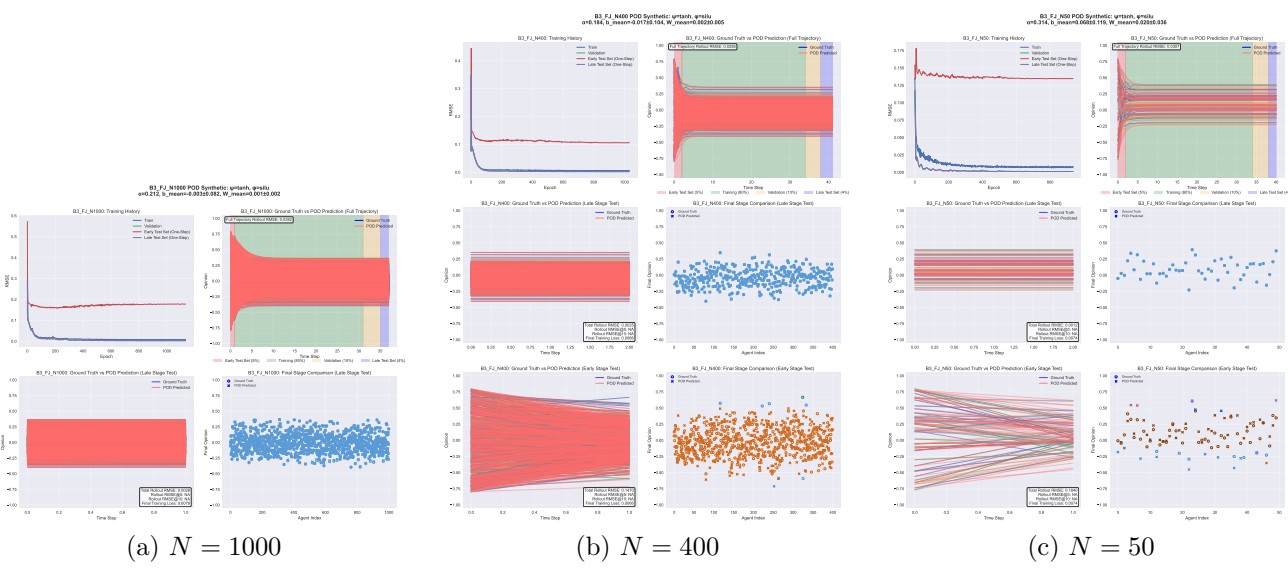

Figure 12: POD learning of Friedkin–Johnsen dynamics

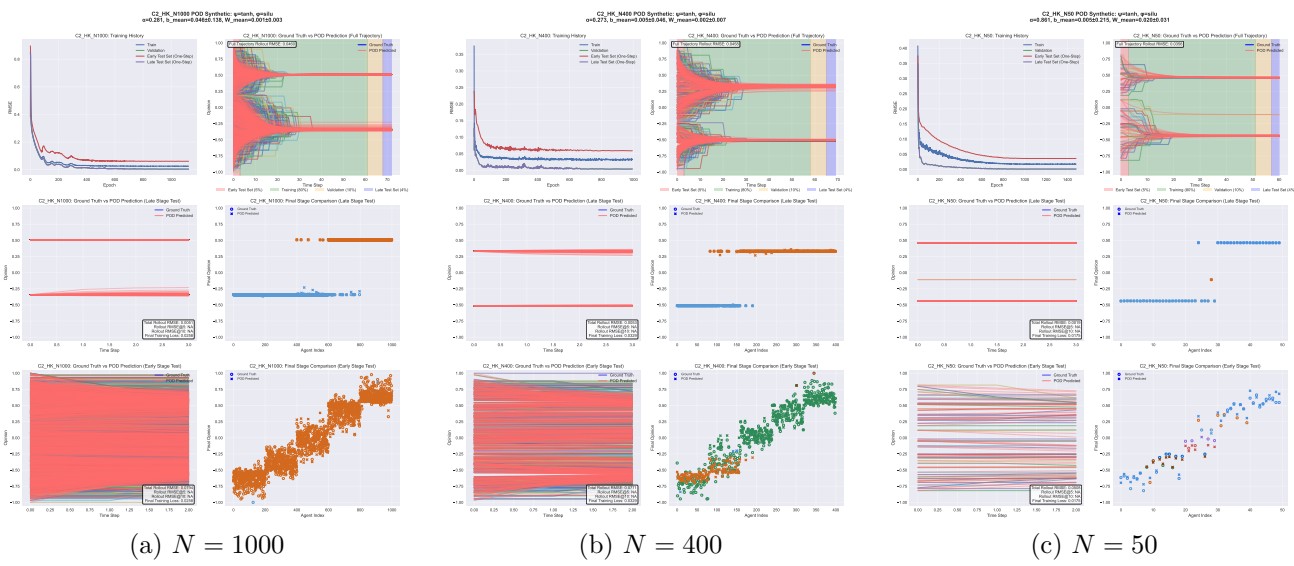

Figure 13: POD learning of Hegselmann–Krause dynamics (wide confidence, $\varepsilon = 0.3$)

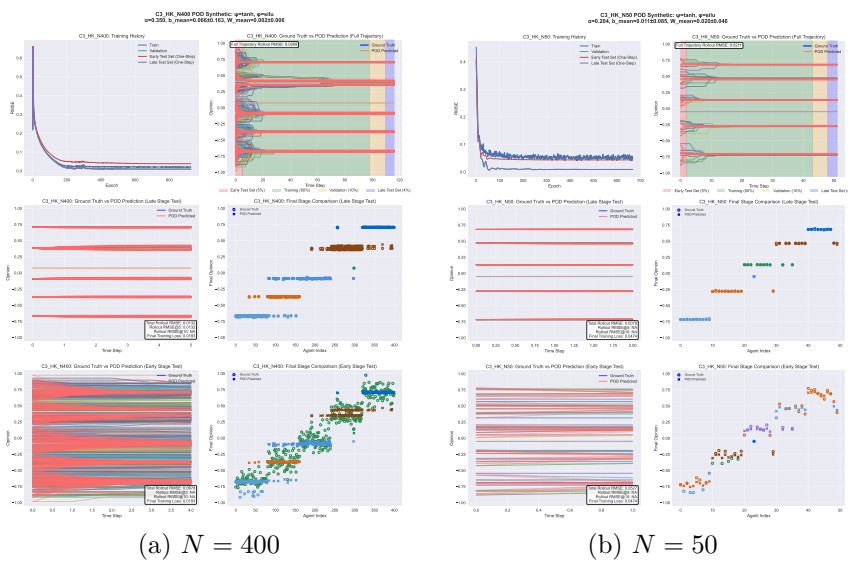

Figure 14: POD learning of Hegselmann–Krause dynamics (narrow confidence, $\varepsilon = 0.12$)

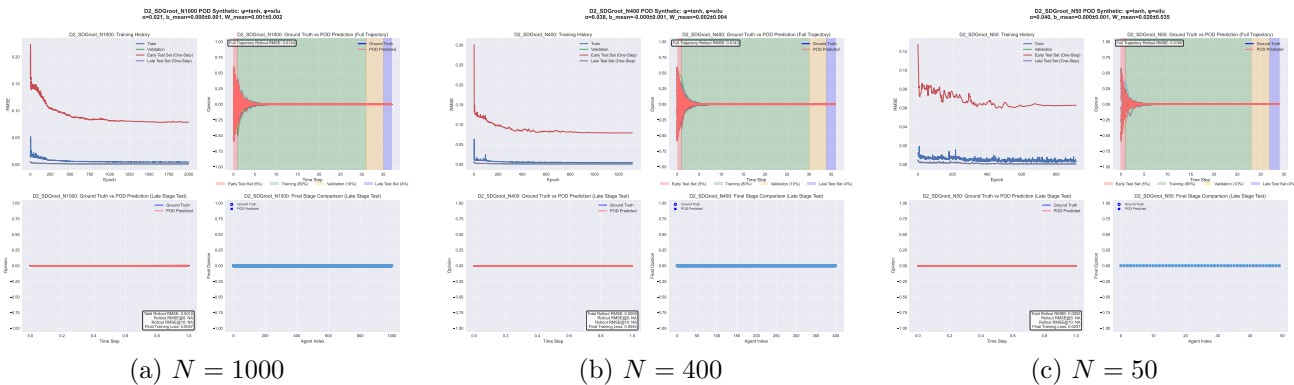

(a) $N = 1000$     (b) $N = 400$     (c) $N = 50$

Figure 15: POD learning of signed DeGroot dynamics (two communities)

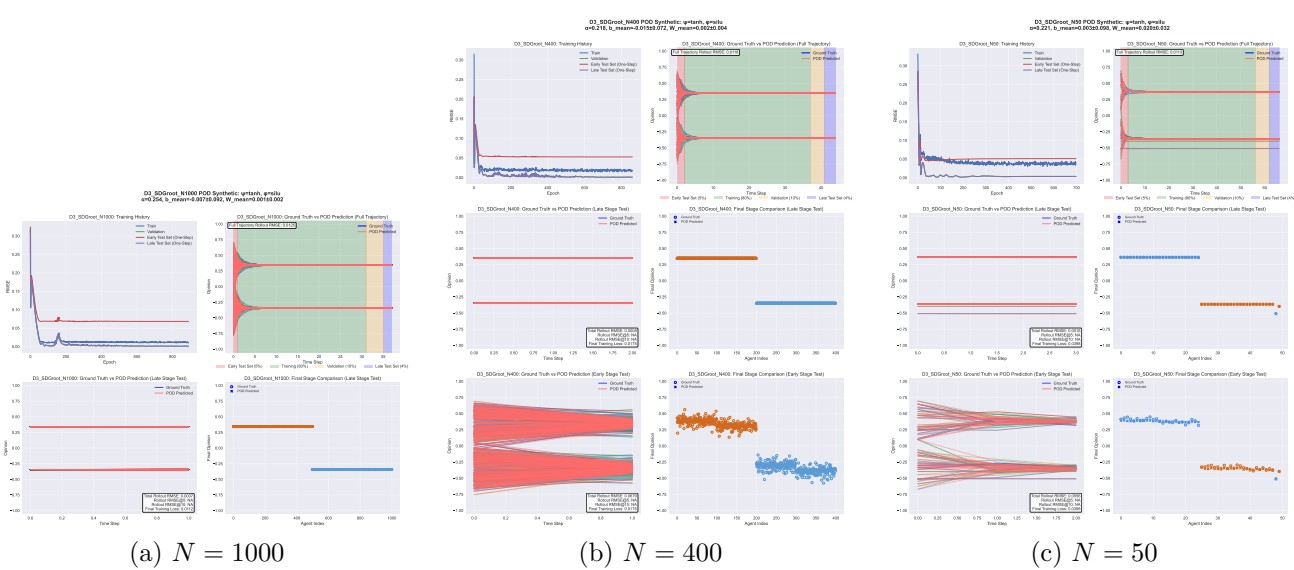

(a) $N = 1000$     (b) $N = 400$     (c) $N = 50$

Figure 16: POD learning of signed DeGroot dynamics (four communities)

