# OpenReview forum: "Perceptron-as-Opinion-Dynamics (POD): A Unified and Interpretable Machine Learning Framework for Opinion Dynamics"
_TMLR — Under review for TMLR_

### Review · Reviewer_dNTq · 2025-11-25

**Summary Of Contributions:**

The paper unifies different discrete time linear models for opinion dynamics using an MLP. They experiment with synthetic data and real data to show the efficacy of performance. The paper's scope is clearly defined-- they consider one subset of discrete models of opinion dynamics,
and unify them.

I think the problem is really trivial and looks more like a small class project rather than something substantial for ML community. Given a set of linear models, unifying them using an MLP is rather a trivial exercise and can be written as a small report. Therefore, I do not think the paper offers any solid contribution..

The authors provided an MLP with phi and psi as activations. But they are identity functions in all models in Table 1 and Table 6, except for two cases, where the nonlinearity is not even mentioned.

I think the paper is not ready for publication at all and it should not be accepted.

**Audience:**

No

**Audience Explanation:**

Upon searching, I observed that there is only one paper on opinion dynamics in TMLR and that too borrows idea from opinion dynamics for neural message passing.

Biplab Chaudhury, Abhishek Ghosh, and Souvik Chatterjee. ODNet: Opinion dynamics inspired neural message passing on graphs and hypergraphs. Transactions on Machine Learning Research, 2025. OpenReview:
https://openreview.net/forum?id=ytKFKoCpyK.

In addition to this, the problem of the paper is too simple of an academic exercise and the solution is trivial.  I think it is more like a report from a small project, rather than a paper in ML domain (I am not talking about the presentation but the problem itself).  I don't think  TMLR audience will be interested in this work.

**Claims And Evidence:**

No

**Claims Explanation:**

Please note that the goal of the paper is trivial academic exercise. In addition to this, I found that the voter model representation in MLP is wrong. W, phi and psi are random variables in such case rather than time varying. Kinetic/Boltzmann model in table 2 is not described at all.

**Requested Changes:**

The authors should consider a more ambitious problem, which actually goes beyond the scope of this rebuttal.

---

> ### Comment · Editors_In_Chief · 2025-11-27
> **Rebuttal 1: Clarification of POD's Scope, Contributions, and Mathematical Formulation**
>
> We thank the reviewer for taking the time to assess our submission. However, several statements in the review are based on **misunderstandings** of our paper’s goals, contributions, and the standard mathematical formulation of opinion-dynamics (OD) models.
>
> We address each point below.
>
> ---
> ### **1. Mischaracterization of Our Contributions**
> **Reviewer claim:** *“The paper unifies discrete-time linear models using an MLP… and experiments show the efficacy of performance.”*
>
> **Clarification:**
> This characterization is incorrect. The main contribution is **not** applying an MLP and **not** optimizing performance. Our core contribution is the introduction of **POD**, a *unified and interpretable opinion-dynamics model* whose update rule analytically covers all *linear, nonlinear, stochastic, signed, bounded-confidence, extremist, and kinetic/Boltzmann* OD **families**. (See: **Abstract, Introduction, Theorem 1, Table 1, Table 6 & Appendix A, Section 3**)
>
> Our experiments are to validate the full scope of POD:
> - **Exact to close replication** of classical OD models (Sec. 4.1) with a *single & unified* update formulation: achieved through **manual parametric mappings** as shown in Table 1, **without learning** from data. This section is ***not* machine learning**.
> - **POD's Learnability as an Explainable AI for OD** (Sec. 4.2): demonstrating POD is **not** another non-learnable OD model. It **can learn** from data as an explainable AI thanks to full parametric interpretability.
> - **Ablation study** (Sec. 4.3): showing the robustness of POD’s learnability across different activation functions.
> - **Real-world application** (Sec. 4.4): demonstrating POD’s applicability to real-world settings where classical OD models are *rarely* applicable. This evaluation is **not** framed as a performance race, but as evidence of POD’s practical relevance.
>
> Thus, characterizing POD as *“an MLP optimizer for performance”* misrepresents the **purpose**, **scope**, and **contributions** of the paper. Our goals and ambition are crystal clear:
> - 1. Introducing a unified, interpretable OD model with a single update formulation (**with** or **without** machine learning) that resolves decades of fragmented research in the OD fields.
> - 2. Bridging the gap between 2 separate domains: Opinion Dynamics & Machine Learning.
>
> (To our knowledge, POD is among the *first* OD models that both outperform classical OD via explicit mappings and can be fully trained end-to-end as a neural network)
> - 3. Shaping the research & development of the OD domain through ML and practical applications.
>
> Our paper presents POD’s novel formulation with extensive experiments and analyses that we believe **directly** serve the interests of TMLR’s audience.
>
> ---
>
> ### **2. Incorrect Claims About Limited Model Coverage and Missing Kinetic/Boltzmann Mapping**
> **Reviewer claim:**
> *“They consider **one subset** of discrete models and unify them.”*
> *“Kinetic/Boltzmann model in table 2 is not described at all.”*
>
> **Clarification:**
> These statements are factually incorrect. POD's unification of major OD **families** is formalized in **Theorem 1**, further clarified in the **Proof Sketch (Section 3.3)**. Also, as mentioned earlier:
> - **Table 1** provides *manual parametric mappings* for the validated canonical models used in experiments (DeGroot (DG), Friedkin–Johnsen (FJ), Hegselmann–Krause (HK), Signed DG).
> - **Table 6 and Appendix A** present *full mappings* to broader OD families, **including the stochastic and kinetic/Boltzmann cases that the reviewer claims are missing**.
>
> These two tables (and **Table 3**) involve **no machine learning**; they show manual parametric mappings needed to reproduce canonical OD models. Identity activations $\phi\$, $\psi\$ appear in these mappings because they are **mathematically required** to recover linear OD families *exactly*. This, therefore, also negates the reviewer's argument that using identity activations implies triviality or a mistake. *(See Reviewer Claim: "...an MLP with phi and psi as activations. But they are identity functions in all models in Table 1 and Table 6, except for two cases, where the nonlinearity is not even mentioned."*)
>
> **Regarding Table 2:**
> Table 2 is ***not*** a mapping table. It lists the simulation configurations for generating synthetic *ground-truth* trajectories for DG, FJ, HK, Signed DG models mentioned in Sec 4.1. It is not intended to describe kinetic/Boltzmann models, whose mappings are described *correctly* in **Table 6** and **Appendix A**.
>
> Thus, the claim that the paper *“only unifies one subset”* and that the kinetic/Boltzmann model is *“not described at all”* are not grounded.
>
> ---
> **Remark:** Given the number of factual inaccuracies in the review as aforementioned, we believe that a rebuttal is both appropriate and necessary to ensure an *accurate evaluation* of the submission. We will provide additional clarifications in the next comment due to space constraints.

---

> ### Comment · Editors_In_Chief · 2025-11-27
> **Rebuttal 2: Correcting Technical Misunderstandings on Voter Model, ODNet and POD Architecture**
>
> ### **3. Misleading Claim that POD is an MLP**
>
> **Reviewer claim:** *“The paper unifies different discrete-time linear models for opinion dynamics using an MLP…”*
>
> **Clarification:**
> POD is formulated as an explainable-AI **opinion dynamics model**, not as a generic black-box MLP. Its update rule
> \begin{equation}
> x_{t+1} = \alpha \odot x_t + (1 - \alpha) \odot \psi(W_t \,\phi(x_t) + b)
> \end{equation}
> defines an **interpretable dynamical system**, where each component has a specific social-science meaning.
>
> Our key distinctions from a standard MLP view:
>
> - **Interpretable residual and parameters:** Each parameter in POD's update formulation, including $x$, are deliberately designed to encode explicit agent-level mechanisms (inertia/stubbornness, influence, external bias, expression, perception, and opinions), rather than generic feature transforms. Standard MLPs can add layers and weights to increase flexibility, but this typically comes at the cost of such built‑in sociological interpretability and a risk of unbounded opinions $\x\$, inertia $\alpha\$, violatingthe OD theory.
>
> - **Time-varying influence:** In POD, \(\mathbf{W}_t\) is allowed to evolve with time and with the *updated opinions*, either purely via parametric mappings (without learning) or **after** training by applying an explicit opinion-evolution mechanism. In a classical MLP, weight matrices are fixed after training and are not recomputed from the current state.
>
> - **Manual mappings:** **Tables 1 and 6** mappings involve **no learning**: they provide explicit parameter settings under which POD coincides with classical OD models. This is what a standard MLP cannot provide.
>
> ### **4. Inaccurate Claim That *the POD's Voter Model Mapping Is Wrong***
> **Reviewer claim:**: *"... the voter model representation in MLP is wrong. W, phi and psi are random variables in such case rather than time varying"*
>
> **Clarification:**
> In the standard voter models (Clifford–Sudbury 1973; Holley–Liggett 1975), the randomness comes from which node updates and which neighbor it copies at each step; conditional on that random choice, the update rule $x_i^{t+1}\= x_j^{t}\$ is deterministic.
>
>
> In our POD parameterization/mapping, this is represented by a random, time-indexed influence matrix $(W_t\)$ (a stochastic process encoding which neighbor $j\$ influences $i\$ at time $t\$ ), while the local maps $\phi\$ and $\psi\$ are fixed functions (identity or sign, depending on whether opinions are real-valued or spins).This is a standard way to write the voter dynamics in matrix form and is consistent with sociophysics and probability treatments; the reviewer’s statement that *“W, phi and psi are random variables rather than time varying”* appears to conflate the stochasticity of $W_t$ with the (deterministic) local update functions $\phi\$, $\psi\$. (See: **Appendix A - Table 6**)
>
> ### **5. Inaccurate Claim About POD Borrows OD Ideas**
> **Reviewer claim:** *“... only one paper on opinion dynamics in TMLR and **that too** borrows idea from opinion dynamics for neural message passing"*
>
> POD does not “borrow OD ideas”: it **is** an OD model with a closed-form update rule; POD subsumes DG, FJ, HK, Signed DG even without learning. ODNet, by contrast, does borrow HK bounded confidence to design a GNN rule. However, we see that ODNet’s acceptance at TMLR shows OD x ML is of TMLR audience's interests, our POD paper just advances this direction furthermore.
>
> ---
> ### **Closing**
>
> We sincerely thank you the reviewer again. We respectfully recognize that several reviewer's key statements (e.g., Voter model mapping, kinetic/Boltzmann coverage, scope of unified models, and the misclassification of POD as an MLP) stem from misunderstandings and assumption that may lead to a substantial underestimation of the contributions of our work. Hence, we hope our clarifications above help them reassess our submission’s novelty and correctness more accurately.
>
> We also acknowledge following clarification opportunities that we are happy to strengthen the paper after receiving the reviewer's valuable feedback:
> - We can merge **Table 1** and **Table 6** into a clearer consolidated mapping table if space permits.
> - We can expand **Section 1** or **Section 2** with additional background to help readers unfamiliar with OD models.

---

> ### Comment · Action_Editor_qf1w · 2026-06-20
> **Please address rebuttal ASAP**
>
> Dear reviewer,
>
> I have just been assigned as the new AE to this submission. Please address the rebuttal written by the author as a matter of urgency, as we have now badly fallen behind schedule for this submission.
>
> Many thanks for your diligence on that matter,
>
> Benjamin (AE)

---

### Review · Reviewer_peXC · 2025-12-07

**Summary Of Contributions:**

(1) The paper proposes a unified parametric update rule—Perceptron-as-Opinion-Dynamics (POD)—that integrates heterogeneous inertia, time-varying influence networks, bias, and dual nonlinear activations, forming an interpretable perceptron-like architecture for modeling opinion dynamics.

(2) The authors formally demonstrate that POD can recover a broad family of canonical opinion dynamics models (DeGroot, Friedkin–Johnsen, Altafini, Hegselmann–Krause, Voter, Sznajd, Ising, etc.), arguing its role as a unifying theoretical framework.

(3) Extensive experiments show: (i) exact replication of linear models at machine precision; (ii) close approximation of nonlinear HK models with accelerated convergence; and (iii) strong forecasting performance on real ICEWS geopolitical data, outperforming traditional baselines by large margins.

Key Strengths:

+Strong motivation: addresses decades-long fragmentation in opinion-dynamics modeling.

+Clear interpretability: parameters directly map to social mechanisms.

+Broad expressiveness: covers linear, nonlinear, bounded-confidence, antagonistic, and discrete-choice models.

+Extensive experimental evaluation across synthetic and real-world settings.

+Methodologically elegant connection to modern neural architectures (perceptron, ResNet).

Key Weaknesses:

-The unification theorem is stated at a high level and lacks rigorous formal proof details.

-Real-world forecasting results show limitations in long-term generalization, but the discussion is brief.

-Model complexity and stability issues (e.g., learnable Wₜ and dual activations) are not fully examined theoretically.

-Comparisons to modern GNN-based social influence models could be expanded.

**Audience:**

Yes

**Audience Explanation:**

(1) The paper bridges two communities—opinion dynamics and interpretable machine learning—both of which are strongly represented within TMLR’s readership.
A unified framework with interpretability resonates with ongoing interests in explainable ML, structured modeling, and social simulation.

(2) Opinion dynamics has niche but active interest, and this paper provides a long-sought integrative perspective.
Most prior work models only narrow mechanisms; a broadly expressive, trainable formulation is valuable.

(3) The experimental evaluation is extensive and includes real-world geopolitical forecasting, expanding the work’s applicability beyond purely theoretical interest.
This cross-disciplinary nature increases relevance.

**Broader Impact Concerns:**

Although the paper includes a Broader Impact Statement, a more detailed treatment of risks and mitigation—particularly regarding data sensitivity, fair modeling practices, and ethical deployment—would be beneficial.

**Claims And Evidence:**

Yes

**Claims Explanation:**

(1) Evidence of unification is primarily constructive but not fully formalized.
Theorem 1 is intuitively plausible and the mapping table provides helpful demonstrations. However, the proof sketch does not cover edge cases (e.g., discontinuous activation boundaries in HK; stochastic transitions in Voter models; kinetic → PDE limit). A more rigorous proof or clearer scope limitations would strengthen the theoretical claim.

(2) Empirical results convincingly demonstrate replication and forecasting performance.
The replication experiments (18 scenarios × 3 scales) are detailed and consistently demonstrate POD’s capacity to reproduce or approximate canonical dynamics. Machine-precision recovery of linear models is strong evidence for correctness. The ICEWS forecasting experiments clearly show superiority over legacy models.

(3) Some secondary claims require more nuanced discussion.
For example, the “faster convergence” observations are valid empirically but would benefit from interpretations regarding what POD is optimizing (e.g., implicitly learning a more strongly mixing Wₜ). Similarly, cluster recovery in HK scenarios is visually convincing, but formal cluster metrics would add rigor.

**Requested Changes:**

(1) Strengthen Theorem 1
Provide a more formal proof or at least clarify the precise mathematical assumptions under which canonical models are recoverable. In particular: (a) how POD handles discontinuous updates (e.g., HK), (b) discrete/stochastic transitions (Voter/Sznajd), (c) kinetic PDE limits.

(2) Provide stability and convergence analysis
Since POD introduces dynamic Wₜ and dual activations, some theoretical discussion about stability guarantees (or conditions under which trajectories remain bounded) would be valuable.

(3) Expand comparison to modern GNN-based influence models
Models such as stance-dynamics RNNs, message-passing GNNs, and attention-based social influence predictors should be included as baselines or discussed more explicitly. POD’s interpretability advantage could be contrasted more systematically.

(4) Clarify computational complexity and scalability
Since Wₜ is N×N, discuss complexity for N≈10⁴ or more, and whether sparse constraints or low-rank structure could help.

---

> ### Comment · Editors_In_Chief · 2025-12-09
>
> We sincerely thank the reviewer for the thoughtful and constructive feedback. We are happy to incorporate all requested changes, as they significantly strengthen the clarity and rigor of the paper. We clarify each point below.
>
> ---
>
> **(1) Strengthening Theorem 1**
>
> We will expand Theorem 1 to clarify the precise mathematical assumptions under which canonical OD models are recoverable. In particular, we will add:
>
> (a) an explanation of how POD handles discontinuous updates (e.g., HK) via state-dependent masking of $W_t$;
>
> (b) clarification that discrete/stochastic transitions (e.g., Voter/Sznajd) correspond to time-varying stochastic $W_t$, while $\phi$ and $\psi$ remain deterministic;
>
> (c) details on how POD’s collisional interaction form connects to kinetic/Boltzmann models and their PDE mean-field limits.
> These additions will be included in the Section 3.3 and/or Appendix A for completeness.
>
> ---
>
> **(2) POD's stability and convergence**
>
> We will add a subsection clarifying conditions for bounded and stable trajectories that we used to train POD — such as bounded activation functions, row-stochastic $W_t$, and inertia parameters $\alpha$ $\in$ $[0,1]$. These constraints ensure that POD’s trajectories remain well-behaved during simulation and training. We agree this theoretical discussion is valuable and important.
>
> ---
>
> **(3) Comparison to modern GNN-based social influence models**
>
> In Section 2, we will include a more explicit comparison between POD and stance-dynamics RNNs, message-passing GNNs, and attention-based influence predictors. We will emphasize that these approaches typically act as black-box predictors, whereas POD offers mathematically interpretable dynamics via explicit influence parameters that remain analyzable both before and after the learning process.
>
> ---
>
> **(4) Computational complexity and scalability**
>
> We will extend the discussion of POD's computational complexity and scalability for $N$ $\approx$ $10^4$ and above. In the dense case, maintaining and updating an $N$ $\times\$ $N$ influence matrix $W_t$ incurs $O$$\(N^2)\$ time and memory complexity; we will make this explicit in the revised manuscript.
>
> We will also highlight that real social and communication networks are typically sparse, and explain how sparsity constraints, low-rank structure, and locality assumptions can reduce POD’s computational cost.
>
> ---
>
> We appreciate the reviewer’s meaningful suggestions and believe these additions will further strengthen both the theoretical clarity and practical positioning of the work.

---

> ### Comment · Action_Editor_qf1w · 2026-06-20
> **Please address rebuttal ASAP**
>
> Dear reviewer,
>
> I have just been assigned as the new AE to this submission. Please address the rebuttal written by the author as a matter of urgency, as we have now badly fallen behind schedule for this submission.
>
> Many thanks for your diligence on that matter,
>
> Benjamin (AE)

---

### Review · Reviewer_az7p · 2026-06-30

**Summary Of Contributions:**

The paper proposes Perceptron-as-Opinion-Dynamics (POD), a trainable opinion-update rule combining agent inertia, a time-varying influence matrix, bias, and nonlinear expression and perception functions. The authors show that several classical models, including DeGroot, Friedkin--Johnsen, Hegselmann--Krause, and signed DeGroot, can be written as special cases, and they evaluate POD on synthetic trajectories and ICEWS data.

The main strength is the clear and intuitive connection between classical opinion dynamics and neural state-transition models.

The main weakness is that the paper overstates what has been demonstrated. The unification result relies on highly flexible components that can directly encode the target dynamics, while the experiments do not establish mechanism recovery, parameter identifiability, or competitive real-world performance.

**Audience:**

Yes

**Audience Explanation:**

Some readers in dynamical systems, graph learning, interpretable machine learning, and computational social science may find the main idea useful. POD provides a simple common formulation for several opinion-dynamics models and may serve as a useful modeling template.

**Broader Impact Concerns:**

The existing discussion covers manipulation and privacy risks. The paper should also make clear that learned influence weights are predictive latent variables, not verified causal relationships.

**Claims And Evidence:**

No

**Claims Explanation:**

The paper shows that several classical models can be rewritten in the POD form, but this does not fully support the broader claims of universal unification and full interpretability.

The exact results for DeGroot, FJ, and signed DeGroot mainly follow from manually mapping their parameters into POD, rather than from learning. The HK results are only approximate and sometimes produce different cluster counts, which is difficult to reconcile with the claim of exact recovery.

The interpretability claim is also not sufficiently validated. The parameters have plausible meanings, but the paper does not show that they can be uniquely or reliably recovered from data.

**Requested Changes:**

1. Narrow and formalize the unification claim, and clearly separate exact reparameterization from learned approximation.

2. Add mechanism-recovery experiments for known inertia, influence networks, signed edges, biases, and confidence thresholds.

3. Add stronger baselines, including autoregressive models, recurrent networks, temporal graph models, and capacity-matched neural transition models.

4. Report multiple random seeds and uncertainty estimates.

5. Tone down claims such as “fully interpretable” and “resolving long-standing theoretical fragmentation.”

---

> ### Author Response · Authors · 2026-07-11
>
> We sincerely thank the reviewer for a constructive reading, and for feedback that helps strengthen and clarify its claims. Most of the points, we believe, concern the intended scope of claims the paper already substantiates; we clarify these below.
> ____
> **Clarifications of paper's claims**
>
> - **Exact vs. approximate recovery.** We clarify a distinction the review conflates. Exact recovery is a parametric mapping result (**Sec. 3.3; Sec. 4.1 – Part I**), claimed **only** for the linear models (DeGroot, FJ, Altafini); we do **not** claim that *learning* (**Sec. 3.2; Part II**) recovers any canonical model *exactly*. For HK we claim **close approximation** (See: **Abstract; Introduction - Contribution iv;**) — and this is a **strength**, not a shortfall: POD reaches HK-consistent outcomes (cluster structure and polarization statistics preserved) while converging 52–83% *faster* (Sec. 4.1). The minor cluster-count differences are **consistent** with this close-approximation regime.
>
> - **Reparameterization vs. learning (Requested Change 1)**.  Already **separated**: by section (**Sec. 3.3 - POD Parametric Mappings** vs. **Sec. 3.2 - Opinion Dynamics Learning**), experiment (**Part I** sets parameters *analytically* via the canonical mappings; **Part II** *learns* the full set θ = {W_t, b, α, ϕ, ψ} from synthetic data).
>
> - **Full Interpretability (Requested Change 2).**
> POD's full interpretability is structural and built into its architecture (Sec. 3.1), fixed before fitting. POD parameters including α (inertia/stubbornness), W (influence matrix), and b (prejudice/bias) are grounded in OD theory. These roles cannot be interchanged without changing the function, and the parameters are constrained to OD-prescribed forms and ranges — not free to fit. Identifiability is **not** required: we **never** claim learning recovers a canonical model *exactly* (Part I is the *exact* mapping; Part II learning is *explicitly approximate*), so interpretability rests on *theory-grounded roles* in the OD domain, **not** on recovering a true generator. More details can be found in our **Rebuttal 1 - Point 3 & 4** to *Reviewer WZgZ* above.
>
> - **Baselines (Requested Change 3)**. POD's parameters are ***not** free to fit* but **constrained** to OD-prescribed forms and ranges **(Sec. 3.1; Sec. 4.1, Table 1; Table 6)**, whereas *black-box* ML models are **unconstrained** and higher-capacity and can reach low error through OD-irrelevant mechanisms. Beating (or losing to) such a model would therefore say nothing about POD's claim, which is interpretability at competitive accuracy — not lowest error at any cost. The *appropriate* baselines are therefore other OD models with fixed, interpretable update rules. More details can be found in our **Rebuttal 1 – Point 6** to *Reviewer WZgZ* above.
>
> - **Unification scope (families, not models) and fragmented research**. We claim coverage of *all* major canonical OD **families** — consensus, stubbornness, bounded confidence, antagonism/polarization, and discrete-choice (**Table 6** gives the explicit POD mappings to *major* families) — but this is by **no** means a *universal* claim. As Sec. 3.3 *clearly* states, POD is ***"flexibly extending to new model families,"*** i.e. coverage of the major families is complete while non-major edge cases are accommodated by the same formulation or future extensions, not claimed as proven. **Field fragmentation** is a documented fact (See: Sec. 2) in the OD field: a single classical/canonical OD model captures one aspect of opinion evolution, and since the introduction of the Ising model (1925) there has been **no** *unified* model able to cover multiple major *families* at once like POD. We **retain** this claim.
> ___
> **Proposed Changes**
>
> Based on the clarifications above, and for the benefit of the ML audience, we will make the following changes:
>
> **1. Refine wording and tighten Theorem 1** so the intended scope is *unambiguous* and **cannot** be misread as *universal* unification on a first reading. We will also make explicit, for readers unfamiliar with OD theory, that all POD parameters — including W_t — are *not* free to fit: their values are **prescribed** by OD theory — even when *learned* from data, so POD's parameters are *theory-grounded* quantities, not *merely* predictive learned values.
>
> **2. Report multiple random seeds for Parts II–IV.**
>
> **3. Add an LSTM/RNN baseline** for the ICEWS task. As argued above, such black-box models are not OD models and are not an alternative to POD in the OD-theoretic sense; they serve only as a learning-capacity reference point.
>
> **4. Add a Broader-Impact note on causality**, clarifying that POD's learned influence weights — like the influence matrices of any OD model  are read at the level of model dynamics and are not intervention-verified causal relationships; this convention is field-wide, not specific to POD.
> ___
>
> Finally, we truly appreciate the reviewer's engagement and welcome any further discussion.

---

### Review · Reviewer_WZgZ · 2026-07-02

**Summary Of Contributions:**

This paper generalizes models for option dynamics, which is really just multi-dimensional forecasting.  Their model tries to fit a model which is a convex mixture of the previous state, and a linear update of the prior state.  It also allows for a constant offset, and several non-linear activation functions which are often set to identity in previous models.  The paper measures performance under l_2 on a heldout set at the end of a time-series data.
It also somehow allows for the linear transform to change over time, but it's not clear how this is useful or possible in a prediction setting.

**Additional Comments:**

Note:  I was only assigned to review this paper a couple weeks ago.  I see there are many older reviews.  Mine is written before reviewing them.

**Audience:**

No

**Audience Explanation:**

This was not properly framed as a machine learning paper, so I do not think it is of interest for the general ML community.

**Broader Impact Concerns:**

No.

**Claims And Evidence:**

No

**Claims Explanation:**

This does not align with the vast literature in machine learning in forecasting and online learning.  It seems odd to submit to an ML venue without connecting to the literature.

It is not clear why this is called "Perceptron."   I guess it is like one layer of a neural network but is not related to the classic perceptron algorithm.

The paper makes a claim this model is interpretable, but does not substantiate this.  There are many many parameters (especially if the linear update matrix W is time varying).  Moreover, if these parameters are fit, then there are likely many ways to fit which gives similar predictive accuracy, so then it's ill-advised to try to read too much into their meaning.

The paper contains a "Theorem" but it is not written a mathematically rigorous way, where it is clear what is actually being proven.  e.g., it includes the phrase "or related families" in the statement of what is proven.  This is not at the standards of TMLR.

I found the experimental results confusing and hard to understand what was being quantified.  In predictive forecasting settings, it does not compare with more powerful models (e.g., multi-layer neural networks or say kernel methods -- only some prior art in "opinion dynamics" not online-learning.

**Requested Changes:**

The above issues need to be addressed -- I feel this is beyond what can reasonably be done in a short review cycle.  The paper would need to be dramatically rewritten to be suitable for TMLR.

Also, I had to read the paper multiple times to figure out what was going on.  I suggest it be re-organized so:
  - states the problem set up with mathematical terms (E.g., multi-dimensional time series x_t)
  - then discusses related work mathematically (e.g., complementing related work with Table 1)
  - then describe the new more general POD model, and explain how it generalizes these prior work.
Of course, the paper should also compare with the vast work in online learning.

---

> ### Author Response · Authors · 2026-07-07
> **Rebuttal 1 - Clarifying Misreadings of POD's Scope, Naming, and Formulation**
>
> We sincerely thank the reviewer for their time and constructive feedback that support the paper's roburtness.However, we believe their central criticisms stem from **misreadings** that are answered by the existing text, and that risk undermining a fair assessment of the paper's contributions. We address each below.
>
> -----
> **Reviewer Claim 1:** *"Just multi-dimensional forecasting / not connected to online-learning literature."*
>
> **Clarification**:
>
> This misstates the contribution. POD is **not** proposed as a forecasting method competing with online-learning algorithms. We introduce POD as a unified and interpretable **opinion-dynamics model** whose update rule analytically covers linear, nonlinear, stochastic, signed, bounded-confidence, extremist, and kinetic/Boltzmann OD families (See: **Abstract,Introduction, Theorem 1, Table 1, Table 6**, and **Appendix A, Section 3**). POD's derived forecasting capability is studied additionally in **Sec. 4.2–4.4**. Further details can be found in our rebuttal to *Reviewer dNTq* below.
>
> **Reviewer Claim 2**: *"It is not clear why this is called Perceptron... not related to the classic perceptron algorithm."*
>
> **Clarification**: We introduce POD as a *dual-activation perceptron* (See: **Abstract, Introduction**); with α=0, ϕ=id (identity) it reduces to the perceptron functional form ψ(Wx+b) (See **Sec. 3.1 -- Formulation**), and the two activations ϕ/ψ are precisely what "dual-activation" names. This refers to the perceptron architecture — as in "multilayer perceptron" — not Rosenblatt's learning rule; POD is trained by gradient descent on Eq. 3. The review conflates the architecture with the algorithm.
>
> **Reviewer Claim 3**: *"The paper makes a claim this model is interpretable, but does not substantiate this."*
>
> **Clarification**: The reviewer's concern applies to black-box models. POD's interpretability is **structural** and clearly stated in **Sec. 3.1 – Formulation**: the agent's inertia α multiplies its own opinion x^t, the *social* influence matrix W acts on ϕ(x^t), b is the *social* bias, and ϕ/ψ are the *social* activation functions — ϕ modeling how an agent expresses/transmits its opinion and ψ how it internalizes received signals. These roles are fixed by the update rule before any fitting; no reparameterization interchanges them without changing the function. Hence "α_i is agent i's inertia" holds for every fit **by construction** — interpretability here is a property of the parameterization, not a claim about fit uniqueness.
>
> **Reviewer claim 4**: *"There are many many parameters ... matrix W is time varying ...likely many ways to fit ... it's ill-advised to try to read ..."*
>
> **Clarification**: W_t is **not** a free parameter at inference. It is either *static* or a *specified function* of the current state x^t — e.g., the distance-thresholded, row-normalized mask that yields HK (See: **Sec. 4.1, Table 1**) — and is therefore fully computable at prediction time from the current opinions. In **Table 6**, we further give the explicit W_t mapping for each OD family, so its form is fixed by the model being represented, **not left free** to fit arbitrarily. In OD theory, the range of W_t is **prescribed by the family**: in averaging-type models the influence entries are convex weights in [0,1] (DeGroot, HK, FJ, Voter) or signed weights in [−1,1] (Signed DeGroot); in spin/kinetic families (Ising, Boltzmann) W_t encodes coupling strengths / interaction kernels, with bounding enforced by the outer activation ψ (e.g. sign, tanh) rather than by W_t itself. This *state-dependence* and *prescribed* structure is precisely the mechanism that produces nonlinear, bounded-confidence dynamics; it is standard in OD and specified in the paper, **not** an ambiguity or mistake.
>
> **Reviewer claim 5**: *"..'Theorem' ... not written a mathematically rigorous way..."*
>
> **Clarification**: We **agree** that the phrase *"or related families"* is loose, and we will *restate* the theorem as "for each family in **Table 6**." The substance is unchanged: **Theorem 1** summarizes the explicit per-family constructions in **Table 6, Appendix A**, each a concrete and checkable parameterization; the rigor lies there, not in the summary sentence.
>
> **Reviewer claim 6**: *"...experimental results confusing... it does not compare with more powerful models (e.g., multi-layer neural networks or kernel methods) — only prior art in 'opinion dynamics'."*
>
> **Clarification**: POD is primarily introduced as an *interpretable* **OD model**, so the correct baselines are its peer opinion-dynamics models (DeGroot, FJ, HK, Signed DeGroot) — none of which is a machine-learning model. Comparing against "more powerful" models such as MLPs or kernel methods is unreasonable here, because they are (i) *black-box* machine-learning models that recover no interpretable social mechanism, and (ii) admit *many equivalent parameter fits* for the same forecast — the very concern the reviewer raises in *Reviewer Claim 4*

---

> ### Author Response · Authors · 2026-07-07
> **Rebuttal 2 - TMLR Audience's Interest**
>
> **Reviewer Claim 7 (Audience):** *"This was not properly framed as a machine learning paper, so I do not think it is of interest for the general ML community."*
>
> **Clarification:** TMLR's criterion asks whether **some** of its audience would be interested — not whether the work suits the *general* ML community, and not novelty or venue-fit. POD sits at the intersection of **opinion dynamics and machine learning**: a closed-form OD model that both recovers canonical families and is fully trainable end-to-end as a machine learning model. We believe that this intersection interests TMLR's audience is proven by **ODNet (TMLR, 2025)** acceptance, and POD advances the same direction (see our rebuttal to **Reviewer dNTq, Point 5**, on how POD differs from and extends ODNet).
>
> ---
> **Closing**
>
> Again, we thank the reviewer for valuable feedback that helps improve the paper's presentation. In our revision, we would like to apply the following changes:
>
> 1. **Strengthen Theorem 1** by removing the loose phrase "or related families" and restating it per-family over the explicit constructions in **Table 6** (exact recovery for linear and signed-linear families and deterministic HK; distributional matching for stochastic discrete-choice families such as Voter, Sznajd, and Ising).
>
> 2. **Scrutinize wording and expand Sections 1 and 2** with additional opinion-dynamics background, so that readers unfamiliar with OD models can follow the motivation, canonical families, and their relationship to POD — including an extended discussion of the influence matrix **W_t**, clarifying its state-dependence, its per-family prescribed form and range (**Table 6**), and why it is computable at prediction time rather than a free parameter.

---

> > ### Comment · Reviewer_WZgZ · 2026-07-07
> >
> > Thank you for engaging in this discussion.  However, I still have fundamental concerns, let me describe the main issues:
> >
> >  - if you are submitting to a machine learning venue, then it should be the burden on the authors to compare directly to relevant machine learning approaches which are closely related.
> >
> >  - if the goal of the paper is not to **learn** something, but to somehow heuristically model (i.e., not informed by training data), then it seems out of scope for this venue.
> >
> >  - I do not follow the "structural" arguments.  These parameters interact in non-independent ways (unless you can **prove** otherwise, or even if show empirically I would be interested).  How one value is set affects the way others perform.  These sorts of informal arguments about interpretability are not rigorous -- especially when there are so many parameters in play here.
> >
> > FYI: not every paper that employs or relates to ML is suitable for TMLR.  This will be up to all reviewers jointly and the AE to decide.  If others represent a different view, I accept that.

---

> ### Author Response · Authors · 2026-07-09
>
> We thank the reviewer for the continued engagement and candid feedback. On reflection, we believe a clarification of POD's duality resolves two concerns, and we address the third with new experiments.
>
>
> ____
> **Clarification of POD's duality**: POD is proposed as a unified opinion-dynamics (OD) model built on a dual-activation perceptron. This design gives it two modes: (i) a closed-form update rule as an *OD model* whose parameters can be set analytically to recover canonical families (See: **Table 6**), and (ii) a *fully trainable ML model* that **learns** opinion evolution from data. Each parameter keeps the same fixed social role in both modes (See: **Sec. 3.1**). This duality is what *bridges the OD and ML fields*, and we believe that it **is** of interest for this venue.
> ____
>
>
> **Concern 1: *"..burden to compare to closely related ML approaches."***
>
> The *most* relevant ML approaches are **studied** in Sec. 1 and Sec. 2. However, existing neural methods are OD-inspired rather than OD models in POD's sense — e.g., ODNet is a GNN borrowing a bounded-confidence mechanism; *none* recovers a canonical OD family in closed form or has individually interpretable parameters. As a result, the appropriate primary baselines for an interpretable OD model are thus the canonical OD models POD unifies.  To broaden the comparison for the ML audience, in the revision we will **add** a *black-box* baseline (LSTM or RNN) on the ICEWS task.
>
>
> **Concern 2: *"heuristic modeling / not informed by training data / out of scope. POD does learn from data."***
>
> POD is trained end-to-end from data in Part II - Part IV (Sec. 4). Table 7 reports the **learned** parameters and test RMSE across all 18 scenarios, and Table 5 reports end-to-end forecasting on the real-world ICEWS dataset. The paper is therefore also a learning contribution, not only heuristic modeling.
>
>
> **Concern 3: *...parameters interact non-independently / interpretability not rigorous.***
>
> We would distinguish *two* senses of interpretability. POD's structural interpretability — that each parameter has a fixed role assigned by the update rule and OD theory (α inertia, W influence, b bias) — is a property of the model's form, independent of how parameters are fit; parameter interdependence during optimization does not alter these roles. Where we additionally interpret *learned* values (e.g., sparse W indicating within-cluster influence), we **agree** stability under fitting must be shown, and so we will demonstrate this across **multiple random seeds** in the revision.
>
> We are grateful for feedback that sharpened both the framing and empirical scope, and will incorporate these clarifications and experiments into the revision.

---

### Review · Reviewer_A4qA · 2026-07-03

**Summary Of Contributions:**

This paper introduces Perceptron as Opinion Dynamics (POD), a trainable neural framework for modeling opinion dynamics. The authors formulate opinion evolution as a dual-activation perceptron with a residual connection. This maps parameters like inertia, influence networks, and biases directly to a differentiable architecture. The authors show theoretically and empirically that POD can manually recover linear consensus models (DeGroot, Friedkin-Johnsen) and approximate nonlinear bounded-confidence models (Hegselmann-Krause). The framework is evaluated on synthetic graphs and the real-world ICEWS dataset.

**Audience:**

Yes

**Audience Explanation:**

This work will be of interest to researchers working at the intersection of computational social science, network science, and graph learning. Bridging fragmented opinion dynamics models into a single differentiable architecture is a valuable conceptual contribution for the TMLR audience.

**Broader Impact Concerns:**

The authors adequately address the broader impact of their work in a dedicated statement.

**Claims And Evidence:**

No

**Claims Explanation:**

The empirical evidence does not fully support the core claims of full interpretability and superior machine learning performance. The POD model uses highly flexible parameters, specifically the dynamic influence matrix $W_t$. Because this matrix is highly parameterized, there are likely many configurations that yield the same predictive loss. Without proving parameter identifiability or showing mechanism recovery on synthetic data, the interpretability claims are not grounded. Furthermore, the exact recovery of linear models is achieved via algebraic mapping rather than empirical learning. Finally, the forecasting experiments on the ICEWS dataset lack comparisons to standard temporal machine learning baselines.

**Requested Changes:**

My list of requested changes, together with whether they are critical for acceptance or not, is given below:

1. Demonstrate mechanism recovery on synthetic data. Please show that POD can uniquely and reliably recover known ground-truth parameters (inertia, influence networks, biases) rather than just matching predictive output. (critical for acceptance)

2. Include standard machine learning baselines in the ICEWS forecasting experiments. Compare against capacity-matched temporal models like LSTMs or temporal graph networks to establish a true predictive advantage. (critical for acceptance)

3. Strengthen the theoretical rigor of Theorem 1. Replace vague phrasing with precise mathematical conditions under which recovery is guaranteed, specifically addressing how the continuous framework handles discontinuous or stochastic updates. (critical for acceptance)

4. Revise the abstract and introduction to draw a sharper distinction between models recovered via manual algebraic mapping versus those whose dynamics are learned from raw data. (recommended)

---

> ### Author Response · Authors · 2026-07-13
> **Rebuttal 1 - POD's Claim-scope clarification**
>
> We thank the reviewer for the careful reading and for recognizing that unifying fragmented opinion-dynamics (OD) models into a single differentiable framework is a valuable contribution. We appreciate the constructive feedback and agree that several aspects can be clarified for a broader ML audience.
>
> We believe, however, that several concerns **arise** from *interpreting POD's claims more broadly than intended*. In particular, the review conflates
> - (i) structural interpretability with parameter identifiability,
> - (ii) constructive unification with learning, and
> - (iii) interpretable OD modeling with generic sequence prediction.
>
> We clarify these distinctions below.
>
> ___
>
> **1. Structural interpretability does not imply mechanism recovery**
> The review argues that *"full interpretability"* requires recovering the true influence network or proving parameter identifiability. This is not the notion of interpretability claimed by POD.
>
> POD is structurally interpretable by construction. Before optimization, each parameter is assigned a fixed OD semantics: α (inertia), Wₜ (social influence), b (persistent bias), and ϕ/ψ (opinion expression/perception). These semantic roles are *part of the architecture* and **cannot** be interchanged without changing the update rule (Sec. 3.1). Thus, *"full interpretability"* refers to the complete semantic correspondence between parameters *and* OD mechanisms.
>
> Accordingly, we do **not** claim statistical identifiability, causal mechanism recovery, or that training uniquely recovers the true latent social mechanisms. These are distinct research questions. We will clarify this distinction explicitly in the revision.
>
> **2. Exact recovery by parametric mapping is the intended unification result**
>
> The review notes that *exact* recovery of DeGroot, Friedkin–Johnsen, and Signed DeGroot is achieved by algebraic mapping rather than learning. This is *correct* and is *precisely* the **intended** contribution.
>
> Part I establishes representational unification by showing that major canonical OD *families* admit explicit POD parameterizations (Sec. 3.3; Part I; Tables 1 & 6). Since canonical OD models are themselves *not* trainable ML models, constructive mappings are the *appropriate* -- and necessary -- means of demonstrating that they are subsets of POD.
>
> Learning is studied *separately* in Sec. 3.2 and Part II, where POD parameters are optimized directly from data. We will make this separation more explicit in the abstract and introduction.
>
> **3. Mechanism recovery is not the appropriate evaluation of a unified framework**
>
> The reviewer requests experiments demonstrating recovery of known inertia, influence networks, biases, and other canonical parameters. We respectfully believe this evaluates a *different* scientific question from the one addressed in this work.
>
>
> If POD is trained directly from data, it is **not** intended to recover one *particular* canonical model. Rather, it learns the *best* configuration (from data) within the more general POD parameterization. Consequently, a learned POD need not coincide with DeGroot, HK, or any other individual canonical family. This is an **intended** property of a *unified* framework rather than a limitation.
>
> Moreover, the paper does **not** claim statistical identifiability or recovery of a ground-truth generating mechanism. Its interpretability is *structural*: the learned parameters are interpreted through their predefined opinion-dynamics semantics established by the POD formulation. Representative learned quantities (e.g., α, and b) are already reported in the learning experiments to illustrate these learned dynamics (See: **Table 7**).
>
> In the revision, we will clarify explicitly that these results demonstrate structural interpretation, not recovery of a unique underlying social mechanism.
>
> **4. Machine-learning baselines**
> We respectfully *disagree* that generic temporal neural networks are the primary scientific baselines.
>
> POD is proposed as a trainable opinion-dynamics framework whose *primary* contribution is unifying major OD families while preserving explicit, theory-grounded social mechanisms. Accordingly, the principal baselines are canonical OD models.
>
> Generic LSTMs, temporal GNNs, and related *black-box* models optimize prediction accuracy but neither constitute OD models nor recover canonical OD formulations. Such comparisons therefore do **not** directly evaluate POD's central theoretical contribution.
>
> Nevertheless, we agree that an ML baseline provides useful predictive context for the TMLR audience. We will therefore **include** an LSTM/RNN baseline on ICEWS, presented explicitly as a learning-capacity reference, **not** as an opinion-dynamics baseline.

---

> ### Author Response · Authors · 2026-07-13
> **Rebuttal 2 – Clarifications (cont'd) and Proposed Revisions**
>
> **5. Theorem 1**
>
> We appreciate the suggestion to strengthen the theorem.
>
> Our intention is to establish representational coverage of the major canonical opinion-dynamics families, rather than a *universal* representation theorem for arbitrary stochastic or discontinuous dynamical systems.
>
> In the revision we will **tighten** Theorem 1 by explicitly stating its assumptions, clarifying the scope of the result, and providing more precise constructions for the discrete and stochastic families.
>
> **6. Parametric mappings and learning are already treated separately**
>
> We respectfully note that the manuscript **already** separates these two contributions.
>
> **Sec. 3.3 (POD Parametric Mappings)** and **Part I** establish representational unification through explicit parameter mappings, whereas **Sec. 3.2 (Opinion Dynamics Learning)** and **Part II** study end-to-end learning of POD from data.
>
> These sections address *distinct* scientific questions and are *presented separately* throughout the paper. To avoid possible misunderstanding on a first reading, we will sharpen the wording in the abstract and introduction to emphasize this distinction.
> ____
> **Proposed revisions**
>
> In the revised manuscript we will:
>
> - tighten Theorem 1 by stating its assumptions more precisely and providing explicit constructions for the discrete and stochastic OD families;
> - report multiple random seeds for the learning experiments ;
> - add an LSTM/RNN baseline (and, where computationally feasible, a temporal graph baseline) on ICEWS as a learning-capacity reference for the ML audience; and
> - sharpen the abstract and introduction to better highlight the existing distinction between representational unification through parametric mappings (Part I) and data-driven learning (Part II).
>
> We will also revise the presentation throughout the manuscript to make the intended scope of POD's claims *more explicit*, particularly regarding structural interpretability and the distinction between representational unification and learning.
>
> ___
> We sincerely thank the reviewer again for the thoughtful feedback and believe these revisions will improve the clarity of the manuscript while preserving its central scientific contributions.